# A Review of SATis5: Perspectives on Commercial and Defense 5G SATCOM Integration

Tien M. Nguyen [1,*], Khanh D. Pham [2], John Nguyen [3], Genshe Chen [4], Charles H. Lee [1] and Sam Behseta [1]

[1] Center for Computational and Applied Mathematics (CCAM), California State University in Fullerton (CSUF), Fullerton, CA 92831, USA; charleshlee@fullerton.edu (C.H.L.); sbehseta@fullerton.edu (S.B.)

[2] Air Force Research Laboratory, Space Vehicles Directorate, Kirtland Air Force Base, NM 87117, USA; khanh.pham.1@us.af.mil

[3] JDTNguyen Consulting Services—Also with Intelligent Fusion Technology, Inc. (IFT), Yorba Linda, CA 92886, USA; johndncva@gmail.com

[4] Intelligent Fusion Technology, Inc., German Town, MD 20874, USA; gchen@intfusiontech.com

[*] Correspondence: tmnguyen57@fullerton.edu

**Abstract:** This review provides a comprehensive review of past and existing works on 5G systems with a laser focus on 5G Satellite Integration (SATis5) for commercial and defense applications. The holistic survey approach is used to gain an in-depth understanding of 5G-Terrestrial Network (5G-TN), 5G-Non-Terrestrial Network (5G-NTN), SATis5 testbeds, and projects along with related SATis5 architectures. Based on the survey results, the review provides (i) outlook perspectives on potential SATis5 architectures for current and future integrated defense and commercial satellite communication (SATCOM) with 5G systems, and (ii) a thorough understanding of problems associated with anticipated outlooks and corresponding studies addressing these problems. The commercial SATis5 architectures discussed here can be extended to civilian SATCOM applications.

**Keywords:** 5G Satellite Integration (SATis5); Commercial SATCOM; Defense SATCOM; Civilian SATCOM; eMBB; mMTC; uRLLC; internet-of-thing; non-terrestrial network; next-generation core network; relay node; single-hop; multiple-hop; communication-on-the-move; cloud; data network

## 1. Introduction

The fifth generation of mobile networks (5G) promises a near-ubiquitous and instantaneous connection for many devices globally. To achieve this promise, the 3rd Generation Partnership Project (3GPP) provided the first complete report on the standardization of 5G wireless technology report in 2018 (Release-15). This release-15 focused on 5G-TN. The next release, Release-16, finalized the first evolution of 5G systems. Currently, 3GPP continues to work on the further evolution of the 5G-TN expanding to 5G-NTN. 3GPP has recently released Release-17 with emphasis on 5G-NTNs. Unlike 5G-TN, the 5G-NTN focuses on networks that involve flying objects such as satellites, high-altitude platforms (HAPs), aircraft, or Unmanned Aerial Systems (UAVs). The HAPs and UAVs are considered by 3GPP as air-to-ground networks. This review focuses on the 5G-NTN scenarios utilizing low Earth orbit (LEO) and geosynchronous Earth orbit (GEO) satellites along with a UAV scenario. The UAV scenario is for improved Intelligence, Surveillance, and Reconnaissance (ISR) missions using 5G and satellite networks.

Since the implementation of 3GPP Release-17 on 5G-NTN, interest surrounding the integration of satellite systems into 5G networks (SATis5) has been growing significantly in both the commercial and defense aerospace industry. This review emphasizes the recent and current efforts in the development and investigation of SATis5 architectures that are fully compatible with 5G-NTN standards for commercial and defense applications. In order to achieve this goal, this review employs a holistic survey approach providing the opportunity to gain an in-depth understanding of recent and current SATis5 works. The survey

emphasizes potential SATis5 architecture outlooks for commercial and defense applications. In addition to the outlooks, the review discusses potential problems associated with the architecture outlook and recommends a set of studies and investigations addressing these problems. Using our holistic survey approach, the review is structured as follows:

- Section 2 presents overviews of 5G-TN and 5G-NTN with emphasis on a high-level understanding of 5G technologies and key 5G system components—These overviews provide some insights into 5G functions related to SATis5 architectures described in the subsequent sections;
- Section 3 captures existing SATis5 survey results with the goal to understand (i) possible commercial and defense SATis5 architectures, (ii) the proposed commercial SATis5 architecture roadmaps, and (iii) associated technical challenges;
- Section 4 presents an overview of recent and existing commercial SATis5 Testbeds and projects available in the public domain;
- Section 5 describes an overview of recent and existing defense SATis5 Testbeds and projects available in the public domain;
- Section 6 discusses and provides our SATis5 architecture outlooks and associated problems and challenges for commercial and defense applications; and
- Section 7 concludes the review with a summary and discussion of our thoughts on the perspectives of SATis5 architectures presented in Section 6. Additionally, this section recommends a list of studies and investigations addressing the problems and challenges identified in Section 6.

## 2. An Overview of 5G Terrestrial and Non-Terrestrial Networks

The overview of 5G-NT and 5G-NTN presented in this section is derived from [1–5]. These references provide a summary of current 5G standards and specifications with an excellent description of the following pertinent 5G technology features:

- Extreme broadband speed using "enhanced mobile broadband" (eMBB) and millimeter-wave technologies'
- Massive multiple-input and multiple-output antenna array and state-of-the-art beam-forming technologies'
- "Massive machine-type communications" (mMTC) for consumer and industrial Internet-of-Thing (IoT), and industry 4.0 "mission-critical machine-to-machine" (MC-M2M)'
- "Ultra-reliable and ultralow latency communications" (uRLLC) for "vehicle-to-vehicle" (V2V) and "vehicle-to-infrastructure" (V2I) communications and autonomous driving'
- Non-terrestrial network (NTN) for communication systems that are to be integrated into the 5G systems and networks, including satellites, UAVs, and/or HAPs. This review focuses on the 5G-NTN using GEO/LEO satellites. An overview of current work on "IoT and UAV integration in 5G hybrid terrestrial-and-satellite networks" is also provided in Section 4.

Two key technological advances in 5G networks are the "virtualization of network functions" (VNFs) and the use of end-to-end "5G Slices" technologies (see Section 4.6). The VNF and 5G Slices technologies allow the 5G network to provide services with a specified quality of service (QoS). As indicated in [2], at a high level, the newly developed 5G network incorporated these technologies has the following three key components:

- gNB: is defined as gNodeB, which is the 3GPP standard terminology for a 5G wireless base station that transmits and receives communications between the user equipment (UE) and the mobile network;
- NGC: also known as Next-Generation Core Network (NGCN), which is the 3GPP standard terminology for 5G wireless next-generation core network. As defined by 3GPP, the NGC is part of the 5G network providing services to mobile subscribers through the "radio access network" (RAN). It is also the gateway to other networks, for instance to the public switched telephone or to public clouds;

- RAN: RAN is the 3GPP standard terminology that is defined as a network that is part of the 5G network that connects (i) UE to other parts of a mobile network via a radio connection, and (ii) UE to the core network.

Figure 1 depicts an enhanced 5G-TN architecture for multicasting and broadcasting services [4]. The CDN and Nx (e.g., N1, N2, N3, . . . , Nx) shown in Figure 1 is the Content Delivery Network (or content provider such as the Internet) and the Nx is the Xth interface specified in 3GPP interface specifications, respectively.

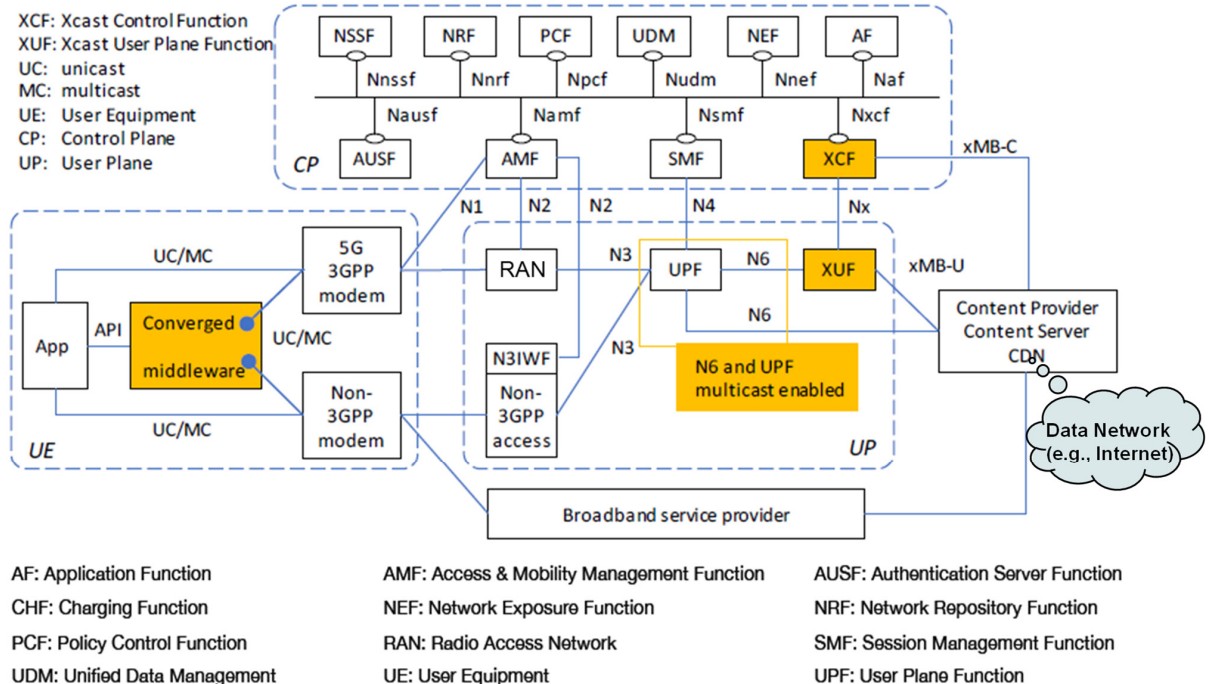

**Figure 1.** Enhanced 5G System Architecture for Terrestrial Multicasting and Broadcasting Network [4].

The 5G-NTN standard specification allows for integrating non-terrestrial networks such as satellites, UAVs, or/and HAPs. This review focuses on the integration of satellite systems, such as GEO and LEO satellites, into 5G terrestrial cellular networks overcoming the range limitation of existing cellular networks. This 5G-NTN architecture is also defined as SATis5 architecture allowing GEO/LEO satellite systems to expand their current services beyond satellite TV, satellite communications, and internet services. Reference [5] discusses 5G-NTN roles addressing (i) the last-mile problem in the terrestrial 5G networks, (ii) coverages for communications-on-the-move (COTM) for moving user-terminals over a large area such as ships at sea or cars driving across the US, and (iii) potential backup of the terrestrial network improving 5G network resiliency.

## 3. Survey of Existing 5G Satellite Integration (SATis5) and Associated Challenges

A survey on 5G-Non-Traditional Network (5G-NTN) focusing on SATis5 architectures and associated challenges was conducted. Based on the survey results [6–12], this section provides an overview of existing SATis5 architectures, the current proposed SATis5 roadmaps, and associated technical challenges. Section 3.1 presents an overview of existing SATis5 architecture options and provides references related to system interfaces and standards. Section 3.2 describes an overview of SATis5 roadmaps. Section 3.3 provides a summary of technical challenges associated with existing SATis5 architecture options.

### 3.1. Overview of SATis5 Architectures

Reference [6] provides a good survey of SATCOM in the new space era where 5G communication networks provide seamless integration between the 5G-TN and 5G-NTN.

The 5G-NTN includes satellites in GEO, MEO, LEO, and very LEO (VLEO), along with both HAPs and "low altitude platforms" (LAPs). Reference [6] has also defined three SATis5 use cases, namely:

- eMBB Satellite Use Case:
  - ○ Backhauling and Tower Feed (BATF): As defined in [6], the satellite provides a matching role by backhauling the traffic load from the edge of the 5G network or broadcasting the popular content to the edge. This matching role optimizes the overall operation of the 5G network infrastructure.
  - ○ Trucking and Head-End Feed (THEF): This use case allows a satellite to directly connect to 5G UEs in remote areas where terrestrial infrastructure is not available.
  - ○ Hybrid Multiplay (HYMP): This use case employs a satellite system to allow 5G service into home/office premises in underserved areas using the proposed 3GPP hybrid terrestrial-satellite broadband connections.
  - ○ Communications on-the-Move (COTM): For this use case, a satellite system is used to provide (i) direct connectivity to COTM platforms (e.g., aircraft, UAV, vehicles (boats, etc.) or (ii) complementary connectivity to COTM platforms supporting 5G services.
- mMTC Satellite Use Case:
  - ○ Wide area IoT services: IoT devices distributed over a wide area and reporting information to or controlled by a central server. Typical SATCOM applications include:
    - ■ Government: Monitoring of oil/gas pipeline status, border, Earthquakes, remote road alerts, etc.;
    - ■ Aerospace and Defense: Fleet management, space asset tracking, etc.;
    - ■ Education: Monitoring and tracking student work, faculty and staff management, etc.;
    - ■ Farming/Agriculture: Farm management, livestock management, etc.
  - ○ Local area IoT services: IoT devices are used to collect local data and report to the central server. Typical applications include a smart grid sub-system (advanced metering) or services for onboard moving platforms, e.g., a container onboard a vessel, a truck, or a train.
- uRLLC Satellite Use Case: As indicated in [6], 3GPP for satellite integration, the SATCOM services require 99.99% link availability, lower than 1ms communication link delay, and package error rate (PER) of $10^{-5}$ for communication link reliability. Typical uRLLC satellite applications include autonomous driving, remote surgery, factory automation, etc.

In addition, [6] provides a description of two key SATis5 architecture approaches for integrating satellite(s) with 5G networks using 5G-NTN standards and specifications, namely:

- 5G-NTN SATis5 Architecture with Transparent Satellite Payloads: The Satellite provides direct or non-direct access to UEs on the ground, i.e., connectivity between the satellite and UEs. For non-direct access, the connectivity would be through a base station (BS), which is a Relay Node (RN) on the ground. 3GPP has provided standards and specifications for the RN and associated air interface for the communication link between gNB and the RN. This architecture can support a satellite use case for eMBB. Figure 2a illustrates this architectural approach. This approach uses the new radio (NR) air interfaces between satellite and 5G-UE, and 5G-RN.
- SATis5 Architecture with Regenerative Satellite Payloads: As shown in Figure 2b, the 5G wireless BS' (gNB) functionalities would be performed by the satellite and thus improve round-trip time (RTT) communications significantly. In addition, regenerative satellite payload would also allow for an inter-satellite link (ISL). The ISL can be used to relay information from the satellites to the ground stations for managing the hand-

over mechanism for a large LEO satellite constellation. This architecture option also provides direct or non-direct access to UEs on the ground. Like transparent payload architecture, the access link to UEs is provided by an RN for non-direct access to 5G UEs. In addition to eMBB, this architecture can also support mMTC and some uRRLC satellite use cases. Like transparent satellite payload, this approach also uses the same NR air interfaces between satellite and 5G-UE, and 5G-RN.

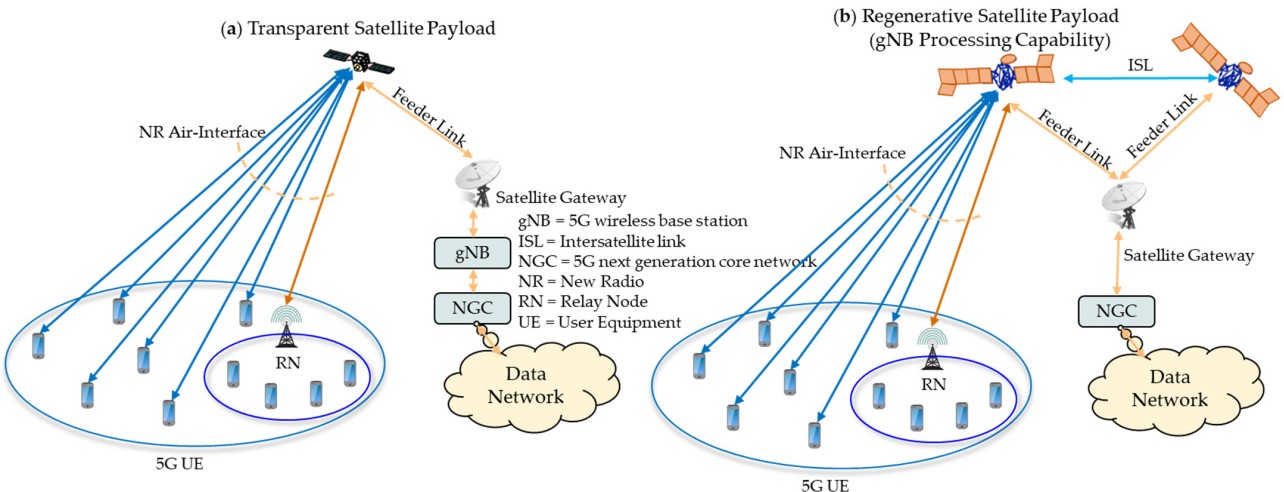

**Figure 2.** SATis5 Architectures for (**a**) Transparent and (**b**) Regenerative Satellite Payloads.

In addition, Reference [6] presents the survey results on a large constellation of LEO satellites or a constellation of MEO satellites interfacing with a private or public central virtual data center (VDC) through a Ground Station Network (GSN). The GSN collects data from satellites and deposits data into a VDC through a public or private Cloud. Routers, servers, and storages are required to transport, route, store, and process the data collected from coming from the GSN. Interested users of the satellite data that want to obtain the data have two options. The first option is to lease the GSN from ground station providers. The second option is to build their own antennas and GSN. The interested users need to access the Cloud (data center) to get the desired satellite data. An example of a typical leased GSN is the Amazon Web Service (AWS) GSN, which is launched by Amazon [7]. The AWS-GSN provides GSN services available today in US-West-2 (Oregon) and US-East-2 (Ohio) to communicate with LEO and MEO satellites. This is performed using either reservation or on-demand scheduling. Figure 3 illustrates a Cloud-based GSN service solution reducing the data access delay.

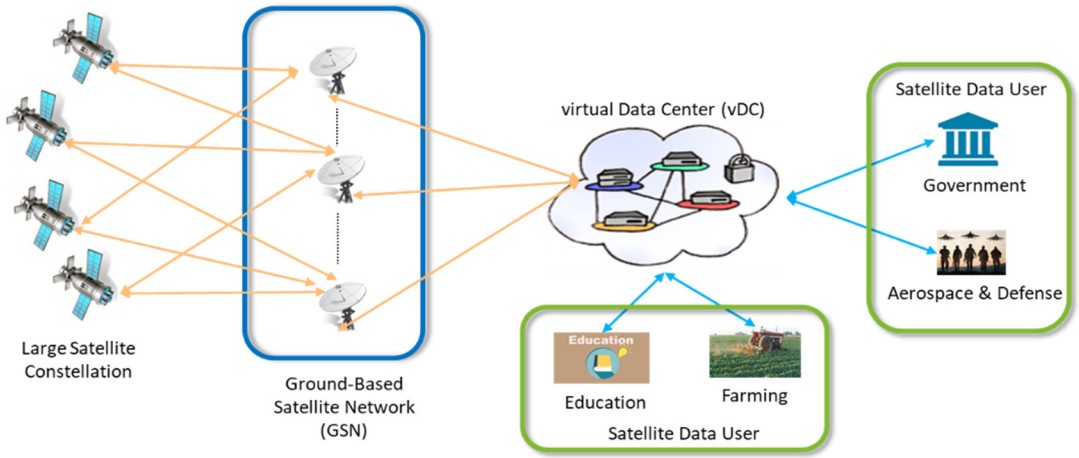

**Figure 3.** Cloud-Based GSN Service Solution for Large Satellite Constellation.

References [8,9] provide an excellent overview of 5G-NTN (also known as 3GPP-NTN) and related SATis5 architectures and associated technical challenges. As pointed out in [8], the "next generation RAN" (NG-RAN) (also known as 5G-RAN) supports the splitting of gNB into two separate units, namely, the "central unit" (CU) and "distributed unit" (DU). As defined in the 3GPP standard and specification, the higher-layer protocol stack for the "New Radio" (NR) (also known as 5G Radio) is divided into two control planes, namely,

- User plane (UP): This plane is for managing data transmission;
- Control Plane (CP): This plane is responsible for the control of the signaling mechanism.

Table 1 provides a list of 3GPP-NTN Technical Reports extracted from [8].

**Table 1.** Related 3GPP-NTN Technical Reports (Extracted from [8]).

| 3GPP Study Item/Objective | Responsible Group | Technical Report | Release |
|---|---|---|---|
| RP-190710: Study on solutions for NR to support NTNs Objective: study a set of necessary features enabling NR support for NTN. | RAN1, RAN2, RAN3 | TR 38.821 | 16 |
| RP-201256: Solutions for NR to support NTNs Objective: specify the enhancements identified for NR NTN with a focus on LEO and GEO and implicit compatibility to support HAP station and air-to-ground scenarios. | RAN1, RAN2, RAN3, RAN4 | N/A | 17 |
| SP-180326: Integration of satellite access in 5G Objective: specify stage 1 requirements. | SA1 | N/A | 17 |
| P-191335: Integration of satellite systems in the 5G architecture Objective: produce normative specifications based on the conclusions identified in TR 23.737 (see below) | SA2 | N/A | 17 |
| SP-181253: Study on architecture aspects for using satellite access in 5G Objective: identify key issues of satellite integration in 5G system architecture and provide solutions for direct satellite access and satellite backhaul. | SA2 | TR 23.737 | 17 |
| CP-202244: Core Network and Terminal (CT) aspects of 5GC architecture for satellite networks Objective of study phase: study the issues related to Public Land Mobile Network (PLMN) selection and propose solutions. Objective of normative phase: support Stage 2 requirements, and satellite access requirements and solutions for PLMN selection | CT1, CT3, CT4 | TR 24.821 | 17 |

As pointed out in 5G-NTN standards and specifications, for NTN, the main impact on the UP comes from the long propagation delays. The delays can cause a potential impact on the "medium access control" (MAC) layer, "radio link control" (RLC) in the Physical (PHY) layer, and packet data convergence protocol (PDCP). The MAC, RLC, and PDCP were modified for 5G-NTN. Reference [8] provides an excellent summary of 3GPP NTN work and related technical reports. Key 3GPP-NTN technical reports related to SATis5 interfaces shown in Figure 2 are provided in Table 1. It should be noted that (i) the technical reports presented in Table 1 were released during 2018–2019, and (ii) only reports related to Figure 2 were extracted from [8].

The impacts of satellite channel characteristics on PHY and MAC layers have been analyzed and presented in [9]. The channel characteristics associated with transmitted waveforms and procedures for eMBB and mMTC-NB-IoT applications were investigated in [9]. [9] also identified and discussed the technical challenges associated with PHY/MAC procedures, including random access, timing advance, and hybrid automatic repeat request. To analyze PHY/MAC layers, [9] has decomposed the two 5G-NTN integration

architectures shown in Figure 2 into four architectures, called A1, A2, A3, and A4. They are defined as follows:

- A1: SATis5 Architecture for Direct User Access Link with Transparent Satellite Payload (PL);
- A2: SATis5 Architecture for Non-Direct User Access with Transparent Satellite PL;
- A3: SATis5 Architecture for Direct User Access Link with Regenerative Satellite PL;
- A4: SATis5 Architecture for Non-Direct User Access with Regenerative Satellite PL.

Table 2 provides a summary of the four possible SATis5 architectures listed above.

**Table 2.** Summary of Four Baseline SATis5 Architectures.

| Payload Type | User Access—Architecture | Satellite Hop | Possible SATis5 Scenario |
|---|---|---|---|
| Transparent | SATis5 A1—Direct Access: SatGW-2-Sat-2-UE | Single | eMBB with GEO Sats |
| | SATis5 A2—Non-Direct Access: SatGW-2-Sat-2-RN-2-UE | Single | mMTC-NB-IoT with Large Constellation (LC) LEO Sats |
| Regenerative | SATis5 A3—Direct Access: A3(a): SatGW-2-Sat-2-UE | Single | eMBB with GEO Sats |
| | A3(b): SatGW-2-Sat-2-Sat-2-UE | Multiple | |
| | SATis5 A4—Non-Direct Access: A4(a): SatGW-2-Sat-2-RN-2-UE | Single | mMTC-NB-IoT and uRLLC with LC-LEO Sats |
| | A4(b): SatGW-2-Sat-2-Sat-2RN-2-UE | Multiple | |

Reference [10] provides an excellent description of four SATis5 architecture options and associated air interface and related standards presented in Table 2. The air-interface and related standards presented in [10] were derived from recent 2018–2019 3GPP-NTN study results. These results are also in the same time frame as the 3GPP-NTN technical reports presented in Table 1. [10] discusses the NR air interface and standards to support 3GPP-NTN with a focus on the above four SATis5 architecture options. Additionally, [10] provides an architecture diagram for transparent and regenerative satellite payloads with single hop and multiple hops. The hops pass through satellite payloads with and without on-board processing for direct and non-direct access to UEs, respectively. Figures 4 and 5 illustrate the concept of single-hop multiple-hop associated with the above four SATis5 options. Figure 4a,b show a single-hop transparent satellite payload for direct access and non-direct access to 5G-UE, respectively. Figure 5a,b describe a single-hop and multiple-hop for regenerative satellite payload for direct access to 5G-UEs, respectively.

As shown in Figure 5b for SATis5 A4 with non-direct access, the generative satellite payload is required to perform either fully or partly gNB functions depending on the development cost trade-off study for on-board processing at the satellite payload vs. ground processing at the RN nodes. A summary of the design features associated with the four SATis5 architectures is derived from [10] and shown in Table 3. Many of the UE air interface and SATis5 Gateway features have been identified in the 3GPP study report. For the LEO satellite use case with a large satellite constellation, their handovers between satellites and 5G UEs are very complicated. The handover events are technically different from the 5G terrestrial access networks. This requires a new study to investigate NGC protocols with various modes of satellite 5G-UE in the SATis5 SATCOM systems ensuring that the NGC can manage the access information efficiently.

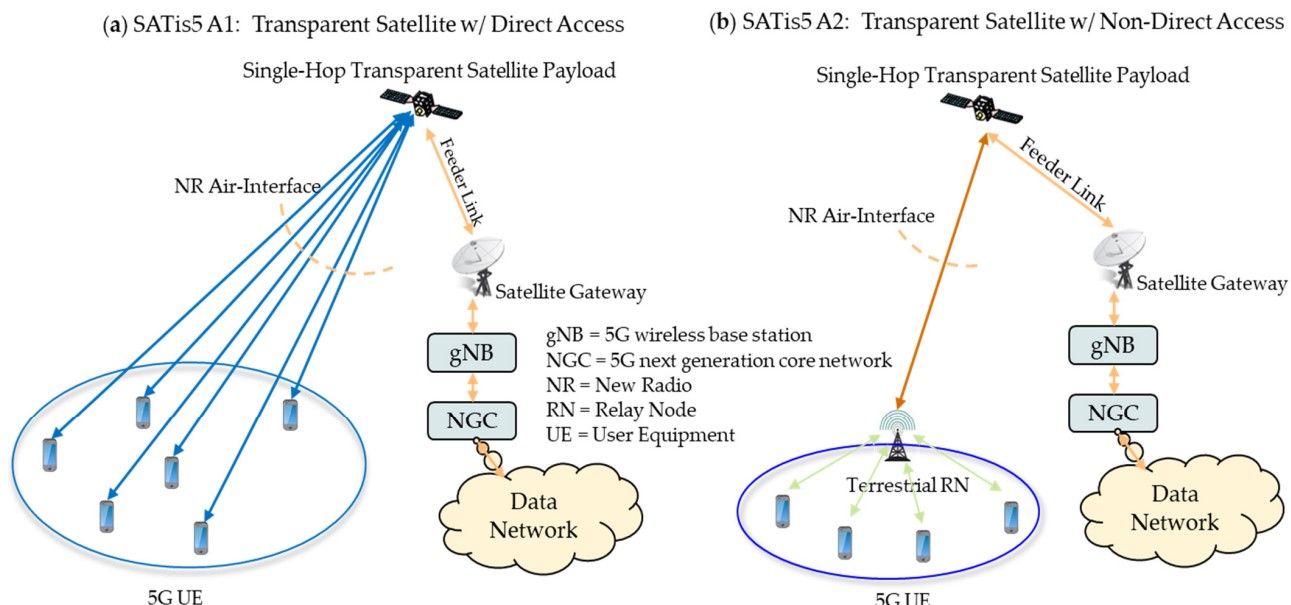

**Figure 4.** SATis5 Single-Hop Transparent Satellite Payload: (**a**) Direct Access SATis5 A1, and (**b**) Non-Direct Access SATis5 A2.

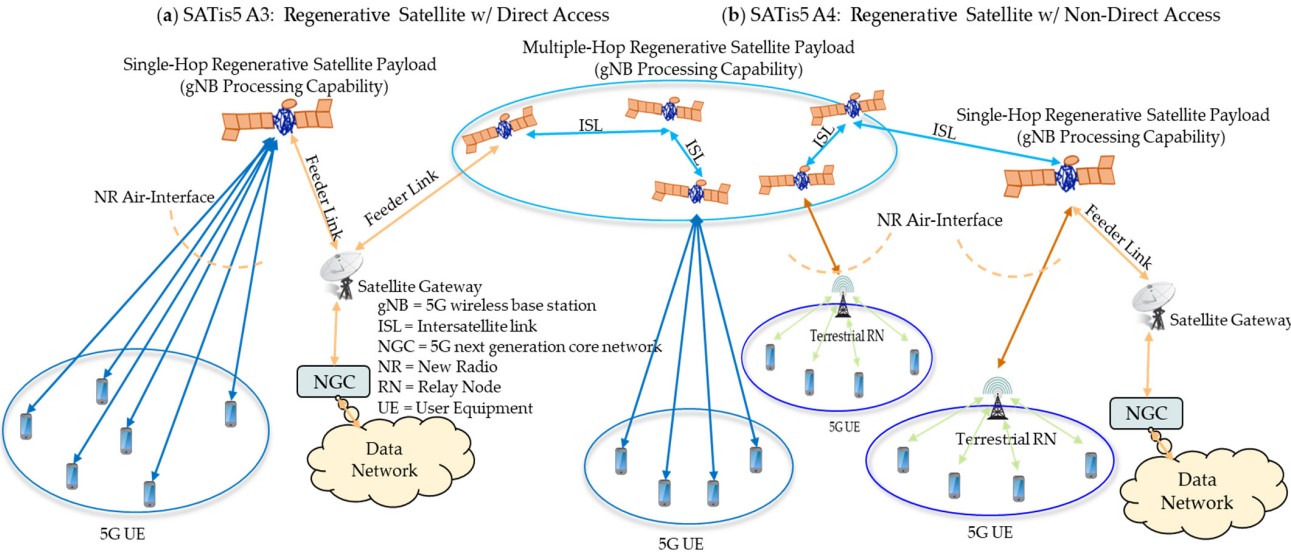

**Figure 5.** SATis5 Single-Hop and Multiple-Hop Generative Satellite Payload: (**a**) Direct Access SATis5 A3, and (**b**) Non-Direct Access SATis5 A4.

**Table 3.** Design Features Associated with Four Baseline SATis5 Architectures.

| SATis5 Architecture | UE Air-Interface Design Feature | Satellite Design Feature | SATis5 Gateway Design Feature |
|---|---|---|---|
| A1: SATis5 with Direct User for Transparent Satellite | UEs access satellites directly | The satellite connects to 5G UEs directly | Satellite Gateway process gNB and NGC protocols |
| A2: SATis5 with Non-Direct User for Transparent Satellite | UEs access satellite via terrestrial RN | Satellite connects to terrestrial RN | |
| A3: SATis5 with Direct User for Generative Satellite | UEs access satellites directly | Satellites process full gNB protocols | Satellite Gateway processes NGC protocols |
| A4: SATis5 with Direct User for Generative Satellite | UEs access satellites via terrestrial RNs | The satellites process part of gNB protocols and the other part can be performed at RN | |

### 3.2. Overview of Current Proposed SATis5 Roadmaps

Ref. [10] analyzes the four SATis5 architectures described in Section 3.1 and proposes roadmaps for 5G integrated commercial SATCOM systems. The analysis is focused on three key SATis5 areas, including:

- Adaptation of Satellite Networks in 5G Architecture: Reference [10] has analyzed the NR air interface design and multi-user transmission for 5G-TN and recommended the adaptation of the terrestrial system in satellite networks;
- Mobility Management, Routing Control, and Load Balance: [10,11] analyzed mobility management, routing control, and load balance associated with intra-satellite, inter-satellite, and inter-access handovers. Based on the analysis, [10] recommended approach for optimizing signaling overhead of handover assuring service continuity;
- Optical-Electric-Mixed Switching Network: [10] conducted research on 5G requirements for multiple service types along with large-scale on-board switching with high rates. [10] also pointed out the need for a switching system that is compatible with different SATCOM and terrestrial links.
- Based on the analysis of the above three key SATis5 areas, [10] has proposed SATis5 roadmaps that include the following three phases:
- Short-Term Phase—SATis5 Serving as Backhaul Links: This phase focuses on the non-direct satellite links architecture. The integrated SATCOM architecture provides backhaul links for short-term 5G-RAN and gNB base stations expanding 5G networks coverage to rural and suburban areas, or emergency services. Figure 6 illustrates a strawman SATis5 architecture for a satellite backhaul network [10]. Section 6 below provides a discussion on the NR air interfaces required at the satellite, gateway, and remote nodes. Note that 5G gNodeB (also known as gNB) at the remote node is very similar to terrestrial gNB located in the metropolitan areas, except that the gNB functions in remote areas are less complex than in metropolitan areas.
- Mid-Term Phase—SATis5 with Satellite Integrated 5G-NGC networks: This intermediate phase supports both direct and non-direct satellite links providing a unified Satellite-5G-NGC network integrated with satellite gateway, 5G-gNB base stations and terrestrial data network. The unified Satellite-5G-NGC automatically selects either satellite or terrestrial networks, depending on the required Quality of Services (QoS) and network availability, providing 5G services for users as required. Figure 7 describes a strawman SATis5 architecture for the intermediate phase. Discussion on the NR air interface is provided in Section 6 below.
- Long-Term Phase—The integration of 5G-UE-Satellite and NR-Satellite Air Interface: The final phase is the full integration of the 5G-UE-Satellite and NR-Satellite air interfaces. As indicated in [10], the satellite and 5G-TN can (i) adopt the same architecture with similar switching techniques and transmission technologies, and (ii) revise existing air interface protocols of the satellite access networks adapting the satellite-terrestrial wireless environment. Note that the 5G terrestrial networks use existing air interface protocols without changing the current 3GPP/5G-UE air interface protocol stack. Discussion of the NR air interface along with an outlook for long-term SATis5 architecture is provided in Section 6.

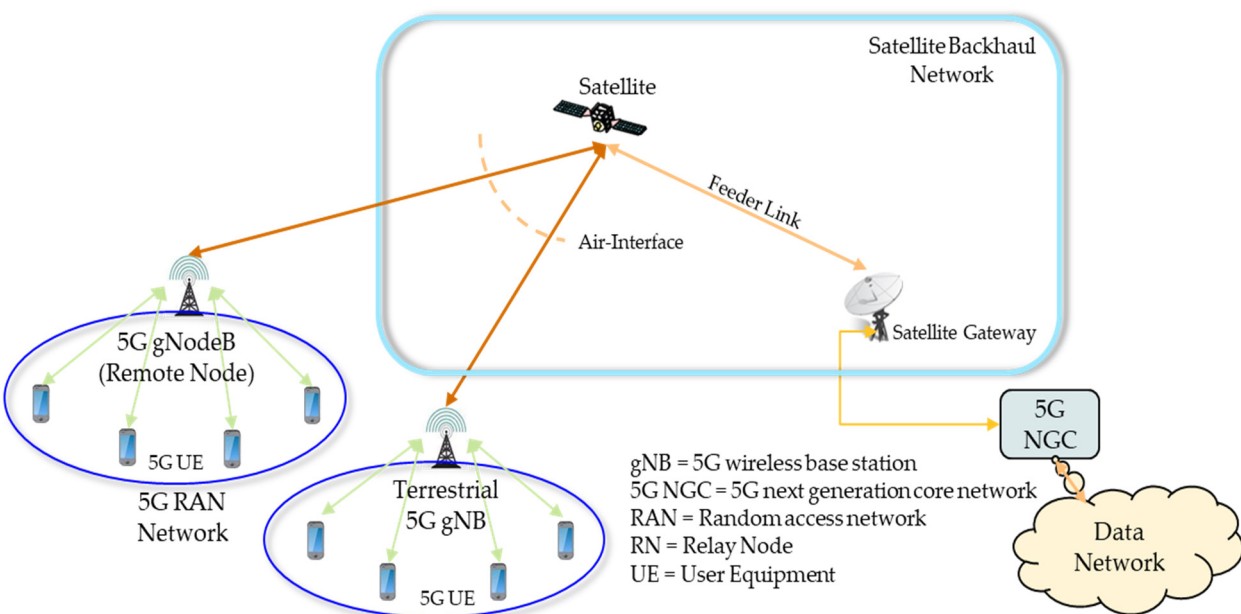

**Figure 6.** Strawman SATis5 Architecture for Short-Term Solution.

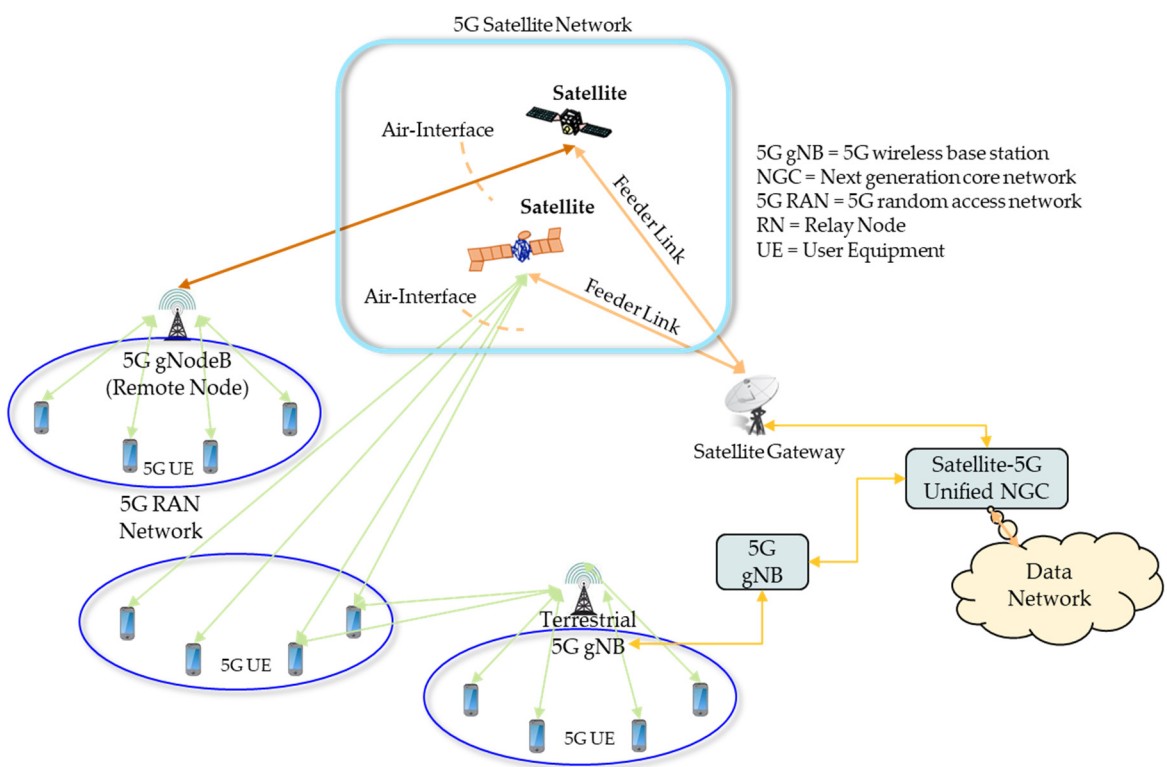

**Figure 7.** Strawman SATis5 Architecture for Mid-Term Solution.

*3.3. SATis5 Technical Challenges*

This section addresses the SATis5 technical challenges associated with the integration of NTN. The NTN consists of GEO/LEO satellites and 5G-NT networks. For integrating GEO satellites with a 5G terrestrial network, the large propagation delay associated with GEO satellites will pose a significant technical challenge in PHY/MAC layers design. The long propagation delay challenge associated with GEO satellites can be mitigated using LEO satellites with a large satellite constellation, i.e., an LEO network with hundreds or

thousands of small satellites providing global coverage, e.g., Starlink being launched by SpaceX [12]. However, LEO satellites pose a new technical challenge associated with large Doppler shifts. The propagation delay associated with LEO satellites is smaller than GEO satellites, but it is still larger than the delay for typical cellular networks. Thus, it still requires some modifications to PHY/MAC protocols for the 5G NR air interface.

Reference [9] has assessed the impacts of channel impairments, including propagation delay and Doppler shift, on PHY/MAC layers associated with SATis5 architectures for A1 and A2 discussed above. The SATis5 system architecture for A1 is assumed for mMTC-NB-IoT services with LEO satellites, and A2 is for eMBB services with GEO satellites. For the eMBB scenario, SATis5 A2 system architecture integrates GEO satellites with 5G networks, and the key technical challenges identified in [9] are:

- Low Signal-to-Noise Ratio (SNR): The low SNR problem inherent with GEO satellites can be mitigated by selecting a power-efficient modulation technique along with high coding gain—The selection of the combined power-efficient modulation-and-coding (ComPEMaC) technique is constrained by the reliability requirements of terrestrial NR systems.
- Forward/Return Link Budget: The link budgets for the GEO satellite feeder and user links must be calculated and optimized to determine the feasibility of the proposed satellite communications links.
- Large Propagation Delay: In practice, a set of ComPEMaC schemes associated with various channel qualities will be available at gNB. A typical gNB selects the most appropriate scheme based on the channel quality indicator reported by the UE. Due to propagation delay associated with the GEO satellite, the channel information at gNB is not updated as required by the 3GPP specification. This delay problem leads to a non-optimum use of the channel resources leading to lower spectral efficiency. To mitigate this long-delay problem, a modification to the 5G "Radio Resource Control" (RRC) procedures is required. For non-direct access, a possible solution is to allow RRC making the RN play an active role between gNB and UE by optimizing both the UE access and the satellite user link air interfaces. This proposed mitigation technique requires an alternative technique for updating channel information between the UE and RN nodes.

Regarding the mMTC-NB-IoT scenario, SATis5 A1 system architecture integrates with large LEO-constellation satellites, and the corresponding key technical challenges are [9]:

- Propagation Delay: Like the eMBB case, RRC procedures are incompatible with SATCOM RTT delays, including:
  - RRC timer procedure;
  - "Random-Access Response" (RAR) time window size;
  - Contention resolution window size;
  - "Timing Advance" (TA);
  - "Hybrid automatic repeat request" (HARQ).
- Large Doppler Effect and Phase Shift: The large Doppler effects and phase impairments associated with LEO satellites can cause interruption to successful transmission. This is because of the 3GPP standardized mMTC-NB-IoT frame structure associated with narrow-band and close OFDM subcarriers. The residual Doppler and Carrier Frequency Offset come from the Doppler compensation and frequency tracking circuitries can cause potential performance degradation.
- Forward/Return Link Budget: LEO satellite power constraint and characteristics of eNB (evolved NodeB), gNB, and narrow-band UE must be incorporated into the link budget calculations of the LEO satellite communication links associated with satellite feeder link and satellite-to-UE links.
1. Battery Life: As specified in 3GPP requirements, the battery life associated with mMTC-NB-IoT-UE is around 10 years. However, for SATis5 services, the battery life is expected to last less than 10 years. This is because the SATis5 devices will require:

       ○      Longer RTT which requires a longer wake-up period time requires performing the access procedures and data transmission;

       ○      Higher transmitted power to close the link.

The commercial satellite industry is currently working on the technical solutions addressing the above challenges and designing the next-generation satellite systems that are integrated seamlessly into the current and future 5G systems. The following section provides an overview of existing SATis5 testbeds and commercial projects addressing SATis5 challenges.

### 4. Overview of Recent and Existing Commercial SATis5 Testbeds and Projects

A survey on recent and existing SATis5 testbeds and commercial projects was conducted using available public sources. Based on the survey results [13–20], this section provides an overview of the ESA ARTES project (Section 4.1), EU-Commissioned SaT5G project (Section 4.2), Starlink Enhancement with 5G (Section 4.3), Avanti Communications and ST Engineering iDirect on SATis5 (Section 4.4), GateHouse SATis5G mMTC-NB-IoT (Section 4.5), NOKIA 5G from Space and Edge Slicing in Next-Generation Virtual Private Network (Section 4.6), and IoT and Unmanned Aerial Vehicle (UAV) Integration in 5G Hybrid Terrestrial-Satellite Networks (Section 4.7).

#### 4.1. ESA ARTES Advanced Technology Project

The "European Space Agency Advanced Research in Telecommunications Systems" (ESA ARTES) project has been addressing the SATis5 challenges. Preliminary results for specific use cases are presented in a whitepaper provided in [13]. There are seven use cases identified in [13] that can be enhanced with SATis5, namely:

- SATis5 Use Case 1: This use case focuses on the improved reliability through alternative connectivity independent from terrestrial, e.g., a 5G core network integrated with a satellite backhaul network operating as a 5G network;
- SATis5 Use Case 2: Provides backhaul for ultra-low latency services deployed at the network edge, e.g., a large amount of video delivery using a satellite backhaul network;
- SATis5 Use Case 3: Provides mobile and nomadic network deployment, e.g., movable connectivity islands with satellite backhaul;
- SATis5 Use Case 4: Provides enhanced and mesh IoT and multimedia services, e.g., mMTC-NB-IoT;
- SATis5 Use Case 5: Provides global coverage use case, e.g., integrated multi-orbit satellite systems providing global service flexibility;
- SATis5 Use Case 6: Provides convergence with data processing and earth observation;
- SATis5 Use Case 7: Provides End-to-end secure services, e.g., governmental use cases.

A SATis5 Practical Proof-of-Concept (PoC) testbed is developed to demonstrate the above seven SATis5 use cases and the PoC results obtained from the testbed are captured in [13]. The SATis5 PoC testbed includes:

- Satellite Ground Systems: Two satellite hubs in Betzdorf and Munich;
- Satellite Ground Nodes: Many satellite-connected nodes are located at various 5G trial locations, including Berlin, Erlangen, Munich, Betzdorf, Killarney, and Nomadic with central nodes located in Berlin and in Betzdorf. Each remote node addresses a different use case;
- 5G Access Networks: LTE, NB-IoT-LTE, and non-3GPP access such as 60 GHz WLAN, WiFi and LoRa. Fraunhofer FOKUS Open5GCore is used to manage the connection of the devices;
- Satellites: The space segment is provided by a commercial satellite operator SES with its ASTRA 2F GEO satellite (28.20E) delivering seamless connectivity between the hub platforms in Betzdorf and Munich and the various 5G trial locations described above.

Table 4 provides a summary of some key SATis5 PoC testbed results related to the above seven use cases.

**Table 4.** A Summary of Some Key SATis5 PoC Testbed Results.

| SATis5 Use case | SATis5 PoC Testbed Use Case Description | Results | Conclusion |
|---|---|---|---|
| Use Case 1 | 5G core network integrated with satellite backhaul network operating as 5G network | Demonstrated capability to leverage COTS 5G core network capabilities to manage a satellite network | Low overhead integration of satellite in existing telecom networks; Open-up satellite Industry to the 3GPP ecosystem. |
| Use Case 2 | Large amount of video delivery using satellite backhaul network | Delivery time delay: - First video—788 ms - Next videos—3 ms | Very high efficiency using the satellite broadcasting and edge caching |
| Use Case 3 | Nomadic node for movable connectivity islands with satellite backhaul | Instant network availability at use case location | Easy to initiate, cost-effective connectivity when and wherever needed |
| Use Case 4 | mMTC-NB-IoT data uplink (U/L) | Non-time critical data U/L: - 287 ms delay - 0.02% packet loss | Highly reliable support for massive IoT |
| Use Case 5 | Integrated multi-orbit satellite systems providing global service flexibility | Demonstrated seamless traffic handover between multi-orbit systems | Integrated multi-orbit satellite systems provide greater service flexibility |

### 4.2. eMBB EU-Commissioned SaT5G Project

The "EU-commissioned Satellite and Terrestrial Networks for 5G" (SaT5G, a.k.a. SATis5) project started in 2017 with a focus on implementing eMBB with satellite networks. Reference [14] has defined four eMBB use cases addressing aspects of these use cases related to extending 5G coverage to full global coverage. The four SaT5G use cases are defined as follows:

- SaT5G Use Case 1: as described in [14], this use case provides "edge delivery" and "offload" for "multimedia content" and "Multi-access Edge Computing" (MEC) "virtual network function" (VNF) software addressing multicast, caching of content to the "edge" and update of 5G network software;
- SaT5G Use Case 2: Provides fixed cell 5G backhaul, i.e., 5G backhauling of remote fixed cell sites;
- SaT5G Use Case 3: Provides 5G to the premises with hybrid multi-play, i.e., delivery of content, e.g., video streaming, using a combination of satellite and terrestrial 5G networks;
- SaT5G Use Case 4: Provides moving platform backhaul, i.e., 5G backhauling from mobile platforms such as ships, aircraft, and land vehicles.

To demonstrate the above four use cases, the SaT5G project has successfully designed and deployed several satellite-based, specific VNFs on "OpenStack and Kubernetes" for incorporation into the SaT5G testbed. This testbed is an integrated SATis5 architecture that allows satellite links to carry "5G network slices" seamlessly. The testbed results for the first three use cases are reported in [14] and an overview of the testbed and associated results are presented in this section. Whereas the final use case was demonstrated in a companion testbed at Zodiac Aerospace in Munich.

A multicast/caching MEC testbed was developed to demonstrate Use Case 1 [14]. This demo showed a multicast adaptive bit rate (mABR) use case based on a 5G MEC-enabled platform for Content Distribution Networks (CDN) integration and efficient edge content delivery via satellite developed by Broadpeak mABR software products and nanoCDN agent. Figure 8 illustrates a SaT5G testbed framework for Use Case 1.

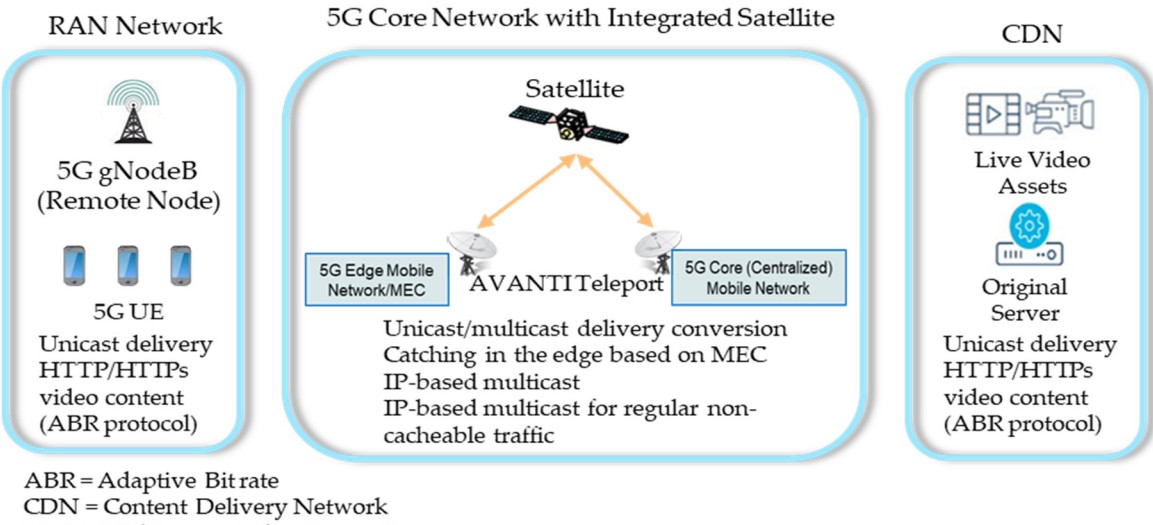

**Figure 8.** SaT5G Architecture for Use Case 1: Multicast/caching MEC testbed.

A multiclient video streaming with a 5G multilinking testbed was designed to demonstrate Use case 2. The testbed provides a CDN that efficiently utilizes the integrated 5G over parallel satellite and terrestrial delivery paths providing enhanced end-to-end (E2E) quality of experience (QoE) for multiple users consuming 4K-layered video. The testbed incorporates mABR adaptation, link selection, and enhanced video streams into a video-segment scheduling network function (VSNF) that is located at the MEC server. The testbed provides a network infrastructure in rural areas where the capacity of terrestrial backhaul is limited. Additionally, the satellite backhaul link offloads the terrestrial backhaul traffic. Figure 9 describes a SaT5G testbed framework for Use Case 2 [14].

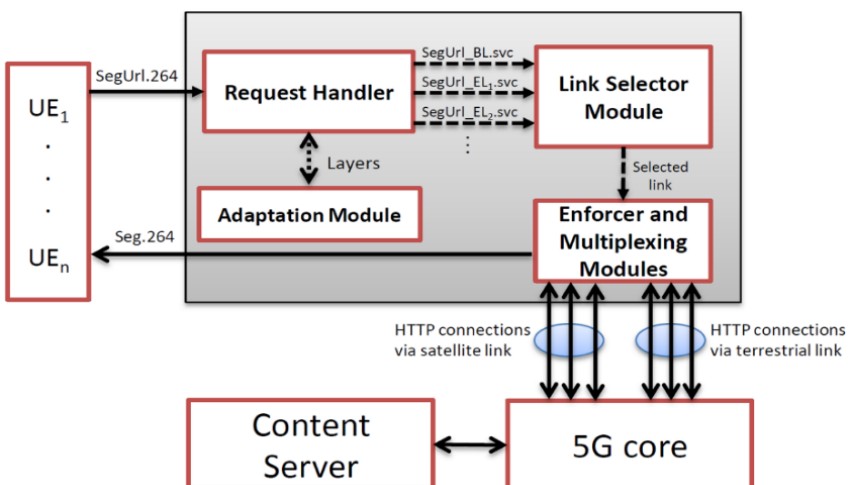

**Figure 9.** SaT5G Architecture for Use Case 2: Multiclient video streaming with multilinking testbed [14].

A 5G hybrid backhauling using a multilinking testbed was built to demonstrate Use Case 3. The testbed focuses on the demonstration of the Multipath version of the Quick UDP Internet Connections (QUIC) protocol (MPQUIC). The software was customized to support hybrid satellite and terrestrial backhauling networks. The 5GIC/QUIC along with MPQUIC were set up in the testbed and test results were obtained and compared with TCP data obtained from the emulated satellite link. Figure 10 depicts a SaT5G testbed

framework for Use Case 3. Table 5 provides a summary of key SaT5G testbed results related to the above three use cases.

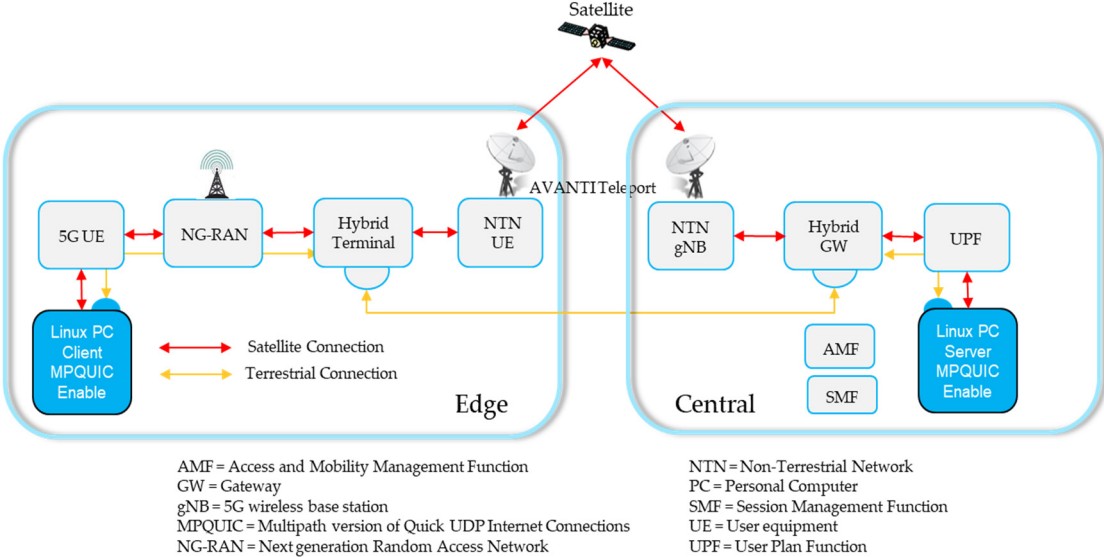

**Figure 10.** SaT5G Architecture for Use Case 3: 5G Hybrid backhauling using multilinking testbed.

**Table 5.** A Summary of Key SaT5G Testbed Results.

| SaT5G Use Case | eMBB SaT5G Testbed Use Case Description | Results | Conclusion |
|---|---|---|---|
| Use Case 1 | 5G MEC-enabled platform for CDN integration and efficient edge content delivery via satellite | (1) Session start-up time–zapping time: 1.5 to 0.9 s; (2) Start-up layer: Always starts on high-quality layer; (3) Bitrate and layers switches: 99% of the session at highest layer; (4) Bandwidth saving: Terrestrial broadband reduced by a factor of 5; (5) Latency: From 12.1 to 3.9 s | (1) Improved video distribution efficiency using mABR over satellite, and video delivery synchronization between screens; (2) Reduced E2E latency using CMAF-CTE Dash over mABR link; (3) Multiscreen capability over satellite; |
| Use Case 2 | Multiclient video streaming with 5G multi linking incorporating mABR adaptation, link selection, and enhanced video streams | (1) Deliver enhanced video quality meeting E2E user QoE; (2) Use of the satellite link considerably reduces the requirements for terrestrial backhaul | Video quality can be improved to meet E2E User QoE by using mABR adaptation and link selection |
| Use Case 3 | 5G hybrid backhauling using multi linking | (1) Performance for short objects is four to 20 times better than satellite alone; (2) Comparable performance for long objects. | Performance can be improved by using multipath protocols to aggregate multilink bandwidths optimally when combined with user QoE for path selection |

### 4.3. Starlink Enhancement with 5G

As of the third of February in 2022, SpaceX has launched 2091 of their expected 42,000 Starlink satellites into orbit. Starlink is a satellite internet network using a large LEO constellation of small satellites operated by SpaceX. Starlink provides advanced satellite internet services that are not currently available with traditional satellite internet services, including video calls, online gaming, streaming, and other global broadband services. Since the Starlink constellation is not fully deployed, there are gaps between satellites causing frequent and short-duration disconnections for global wideband applications that required

continuous connections, e.g., remote telework, video calls, and gaming. In addition to this challenge, Starlink does not currently provide mobile broadband services. To address the Starlink challenges, Simplewan has considered integrating 5G (and existing 4G) into the Starlink network [15]. Simplewan has developed a secondary and complimentary internet connection device to pair the 4G/5G with Starlink service providers. The Simplewan device employs Software-defined wide area networks (SD-WAN) technology enabler allowing the internet service provider to aggregate broadband connections between Starlink and 4G/5G networks automatically.

### 4.4. Avanti Communications and ST Engineering iDirect on SATis5

Avanti Communications and ST Engineering iDirect (AC-STE-iDirect) have played an important role in the EU-Commissioned SaT5G project. AC-STE-iDirect has provided the Avanti Teleport and successfully integrated the Teleport with a 5GIC remote node and G5IC central node using a live commercial AVANTI satellite network [16]. For the SaT5G testbed, the Avanti Teleport serves as a 5G-enabled Intelligent Gateway (5G-iGW) connecting the satellite with the 5G testbed at the University of Surrey Innovation Center (5GIC) remote node and 5GIC central node. For this 5G-NTN use case, the integrated 5G-NTN includes a remote satellite terminal located at a remote node. This configuration allows for the live connection over satellite to the 5G-iGW located at the 5GIC central node. The keys for this configuration are: (i) the satellite connection uses the native satellite radio at the physical layer, (ii) the 5G-iGW provides the "physical network functions" for ending the native satellite connection using a RAN VNF, also known as "virtualized satellite RAN", and (iii) a standard and unmodified commercially available virtualized NGC, also known as "5G core network". Figure 11 describes AC-STE-iDirect Teleport serving as a 5G-GW.

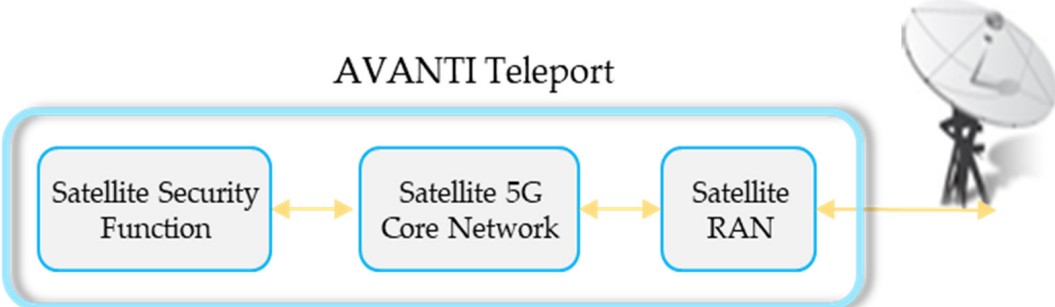

**Figure 11.** AC-STE-iDirect Teleport: 5G-Enable Intelligent Gateway.

### 4.5. GateHouse SATis5G mMTC-NB-IoT

The GateHouse team has been supporting the ESA ARTES project focusing on 5G mMTC-NB-IoT compliance solutions for small satellites with global coverage for maritime, agricultural, critical infrastructure, or oil and gas industry applications [17,18]. Currently, the team is working on low-cost commercial satellite solutions related to the physical layer, MAC layer, and other layers for enabling terrestrial mMTC-NB-IoT for LEO usage. GateHouse is a member of the 3GPP standardization group and actively contributing to the standardization of 5G-NB-IoT-NTN. Their solutions will be fully compliant with the current Release-17 and future releases. The GateHouse SatCom, a subsidiary of GateHouse, is currently collaborating with Addvalue Innovation, a subsidiary of Singapore Exchange Mainboard-listed Addvalue Technologies, to develop the world's first 5G mMTC-NB-IoT satellite communication terminal. The goal is to provide a SATCOM mMTC-NB-IoT terminal that is fully functional with all 5G satellite networks worldwide and compatible with 5G 3GPP standards. The mMTC-NB-IoT user terminals (UT) are ideally multimode IoT devices, capable of using either a terrestrial network or a 5G-NTN network for real-time tracking and monitoring applications, where the user's device moves in and out of terrestrial coverage.

### 4.6. NOKIA 5G from Space and Edge Slicing in Next-Generation Virtual Private Network

Nokia provides its view of 5G from space and the roles of satellites in 5G [19]. Nokia also foresees 5G signals from space complementing 5G terrestrial infrastructure on Earth allowing seamless connectivity to cars, trucks, airplanes, vessels, and IoT devices in remote and rural areas. One of Nokia's latest 5G business and technology innovations is the combination of 5G Virtual Private Network (VPN) service with Edge Slicing and Edge Cloud Applications, also known as 5G Edge Slicing [20]. The 5G Edge Slicing allows for an end-to-end solution that works across 4G/5G RAN, transport layer, 4G/5G NGC, and an enterprise cloud with a distributed cloud core deployed at the enterprise premises or in the proximity of the operator's 5G Edge. Figure 12a illustrates a concept for 5G Edge Slicing deployment for 5G terrestrial network deployment. Figure 12b shows a 4G/5G end-to-end solution for 5G VPN with Nokia Edge Slicing [20]. As mentioned earlier, for 4G networks, the base station is referred to as evolved Node B (also known as eNB or eNodeB).

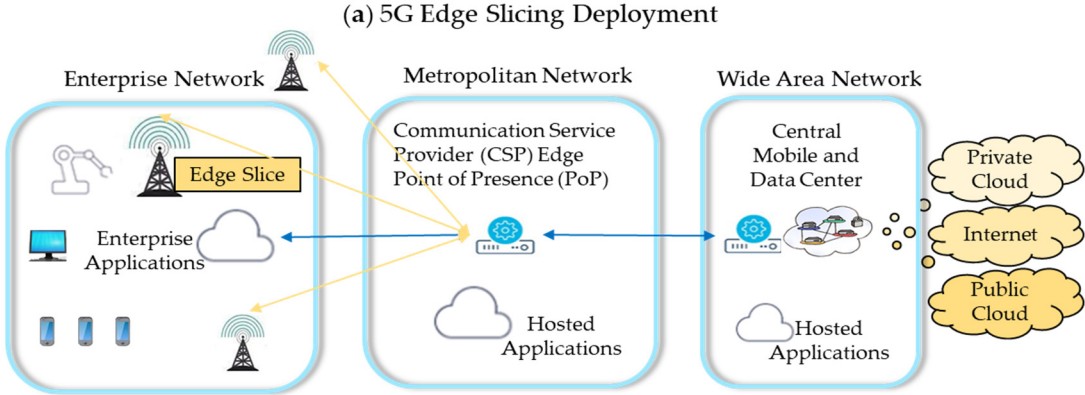

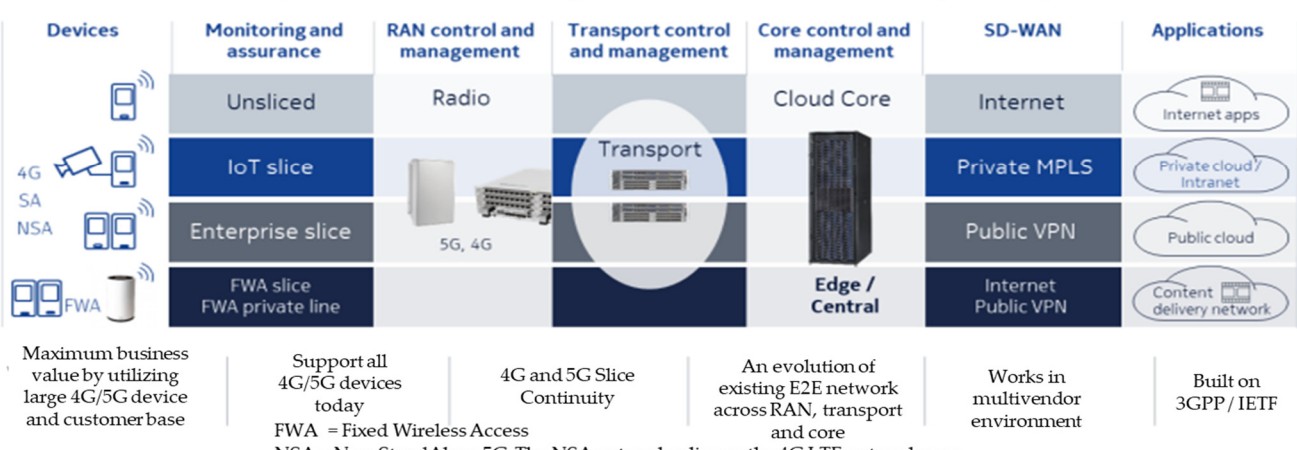

**Figure 12.** 5G Edge Slicing Technology Enabler in 5G VPN: (**a**) 5G Edge Sclicing Deployment, and (**b**) 5G End-to-End Slicing Solution for 5G VPN Edge Slicing [20].

### 4.7. Integration of UAV and Satellites with 5G Network—mMTC-NB-IoT Use Case

Recently, [21] has addressed the IoT environment as UAV integrated with satellite and 5G terrestrial networks. This reference has addressed the joint integration of UAVs and satellite technologies for future 5G networks. The IoT environment includes both short and long-range IoT operations. The short-range IoT includes Bluetooth, Zigbee, and WiFi. The long-range IoT includes LoRaWAN SigFox and Ingenu. The UAV technology can have many applications in practice, including but not limited to homeland security, border surveillance, and goods transportation. When UAV technology is integrated with IoT technology, the integrated technology can be applied to support defense operations,

disaster management, crowd surveillance, real-time road traffic monitoring, earthquake, and environmental disasters monitoring, meteorology management, etc. Reference [21] provides a solution for the integration of the combined mMTC-NB-IoT-and-UAV with satellites and 5G terrestrial networks. Figure 13 shows a top-level architecture for integrating IoT-UAV with satellite and 5G terrestrial networks. The key features associated with this proposed SATis5 architecture are:

- Satellite Layer: Handover mechanisms for both satellites and UAVs are required to manage the connectivity during the handover events—the handover mechanism is required to implement on both satellite and satellite ground stations.
- UAV: Routing protocols are required onboard UAVs for distributing data to IoT devices. In addition, UAV's onboard storage is required to address the loss of connectivity between UAVs or between satellites and satellite gateways caused by outage events.

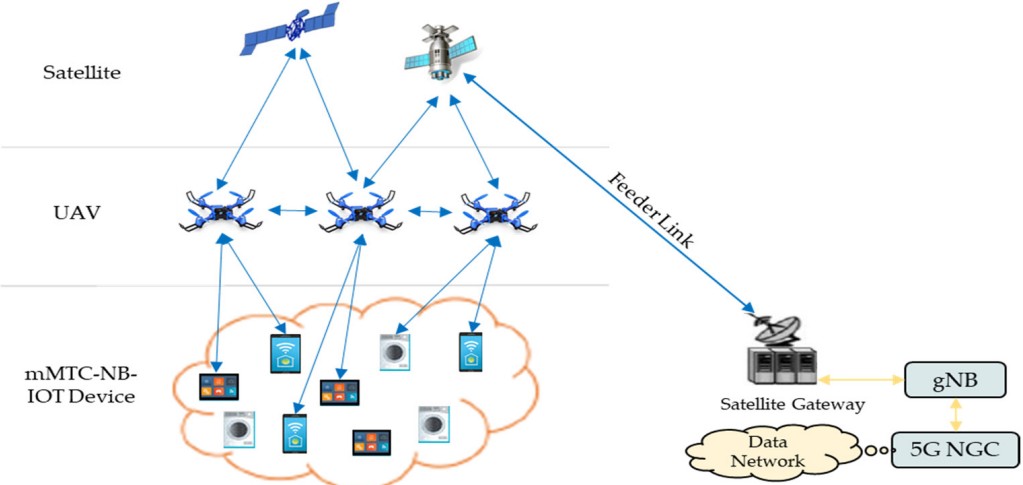

**Figure 13.** Architecture for integrating IoT device-UAV-satellite and 5G terrestrial network.

## 5. Overview of Recent and Existing Defense SATis5 Testbeds and Projects

Currently, works on defense SATis5 architectures are limited and not available in the public domain. Defense SATCOM (also known as MILSATCOM) requirements are different from ordinary SATCOM systems. Defense SATCOM has additional security requirements on the communications links, e.g., transmission security (TRANSEC), emission security, and data encryption/decryption. These security requirements are designed to protect communications links in hostile environments [22]. As described in [22], the defense physical, datalink, and MAC layers for SATCOM terminals are not the same as for civilian and commercial SATCOM terminals. Integrating defense satellites into 5G networks, i.e., defense SATis5 requires addressing the protocols for these layers. The DoD and aerospace defense industry have recognized defense SATis5 integration challenges for multidomain defense operations, such as US DoD Joint All Domain Command and Control (JADC2) program [22–29].

Lockheed Martin's (LM) "5G.MIL" project is currently investigating an approach to integrate MILSATCOM with tactical gateway capabilities and 5G enabling JADC2 across all battlefield assets [24]. LM has also been collaborating with Keysight Technologies on defense SATis5 with a focus on LM-5G.MIL testbed with hosted Keysight's integrated 5G solutions so LM teams can address 5G solutions for multiple defense applications [25]. Keysight integrated 5G solutions include UE emulator and 5G core network and traffic simulator [26]. Ref. [26] also addresses several 5G applications for defense operations, including (i) Integration of global navigation satellite system (GNSS) with 5G networks allowing for 4G/5G UE to establish its position, frequency, and time reference, (ii) 5G communication on-the-move (COTM), and (iii) Security threat for military and government

operations. Additionally, LM also teamed up with Omnispace to jointly develop a global space-based 5G-NTN. Omnispace foresees a "one global network" with 3GPP-compliant "one global network" supporting a variety of civilian, commercial, and defense applications. This includes but is not limited to autonomous vehicles, unmanned aircraft, public safety, and smart agriculture [27].

Recently, Hughes secured a DoD three-year contract to develop a stand-alone SATis5 network located at Whidbey Island Naval Station, Washington [28]. The satellite-enabled 5G wireless network will support the naval air station in terms of operations, maintenance, and flight traffic management. As indicated in [28], the new SATis5 network leverages DISH Wireless and GEO/LEO satellites. For GEO, the SATis5 network uses JUPITER high throughput satellites. For LEO, the network leverages OneWeb LEO.

In 2021, VSAT secured US DoD two three-year research contracts for evaluating the feasibility of 5G connectivity on the battlefield [29]. The contracts focus on two key areas, namely, (i) Research Area 1 focuses on providing C2 hardware packages that bolster C4ISR operations, cybersecurity, and networking software, and (ii) Research Area 2 emphasizes the deployment of secure 5G nodes at the tactical edge. Thus, VSAT will be responsible for testing 5G resiliency in warfighting operations that require large bandwidth applications such as ISR missions.

## 6. Outlook Perspectives on SATis5 for Commercial and Defense Applications

In general, satellite enterprise perspectives can be classified into three categories, namely, commercial SATCOM (also known as COMSATCOM) enterprise, civilian SAT-COM enterprise, and defense SATCOM (also known as MILSATCOM) enterprise. The COMSATCOM and civilian SATCOM enterprises have similar SATCOM requirements and associated characteristics in terms of acquisition, deployment, maintenance, operations, and satellite network management. Thus, perspectives on the commercial SATis5 architectures are expected to be the same as civilian architectures. COMSATCOM SATis5 architectures can be extended to civilian architectures. On the other hand, the defense SATCOM enterprise is quite different as compared to COMSATCOM and civilian SATCOM enterprises. Presently, the US defense acquisition process has been revised to adapt to the latest commercial technologies to combat dynamic threats from adversaries. This section provides outlooks on perspectives for potential SATis5 architectures with emphasis on commercial and defense applications.

The survey results described in Section 4 have provided a perspective on a potential SATis5 architecture for civilian and commercial applications. Section 6.1 below captures this outlook perspective that is agnostic of 5G access and use cases. Likewise, the survey results presented in Section 5 have provided a perspective on potential SATis5 architecture for defense applications. Section 6.2 captures this outlook perspective.

### 6.1. Commercial SATis5 Applications

As discussed in Section 4, civil and commercial applications for current SATis5 architectures for eMBB and mMTC-NB-IoT use cases are of the most interest [13–21]. Recently, Reference [30] provides a good high-level summary of these SATis5 architectures. It focuses on SATis5 architectures that are 5G access agnostic and applicable for both eMBB and nMTC-NB-IoT with either fixed or mobile user equipment and/or terminal. Figure 14 describes a top-level perspective on a potential SATi5 architecture for civilian and commercial applications. This architecture perspective fully complies with ETSI TR 103 611 for scenario A3 [30]. The key interfaces shown in Figure 14 are specified in the 3GPP/5G-NTN technical report (ETSI TR 103 611) as follows [31]:

- N1 interface: This is for UE service, it is implemented between the UE and "Access and Mobility Management Function" (AMF) at the core network through gNB located at satellite customer provided equipment (CPE);
- N2 Interface: This interface is for RAN and UE service, and it is implemented between the gNB and the core AMF;

- N2 Interface: This interface is for the NTN-NT-UE service, and it is implemented between the NTN-LTE-gNB and the core AMF;
- N3 Interface: For NTN-NT-UE service, it is implemented between the NTN-LTE-gNB and the core "User Plane Function" (UPF);
- N3 Interface: For RAN and UE service, it is implemented between the gNB and the core UPF.

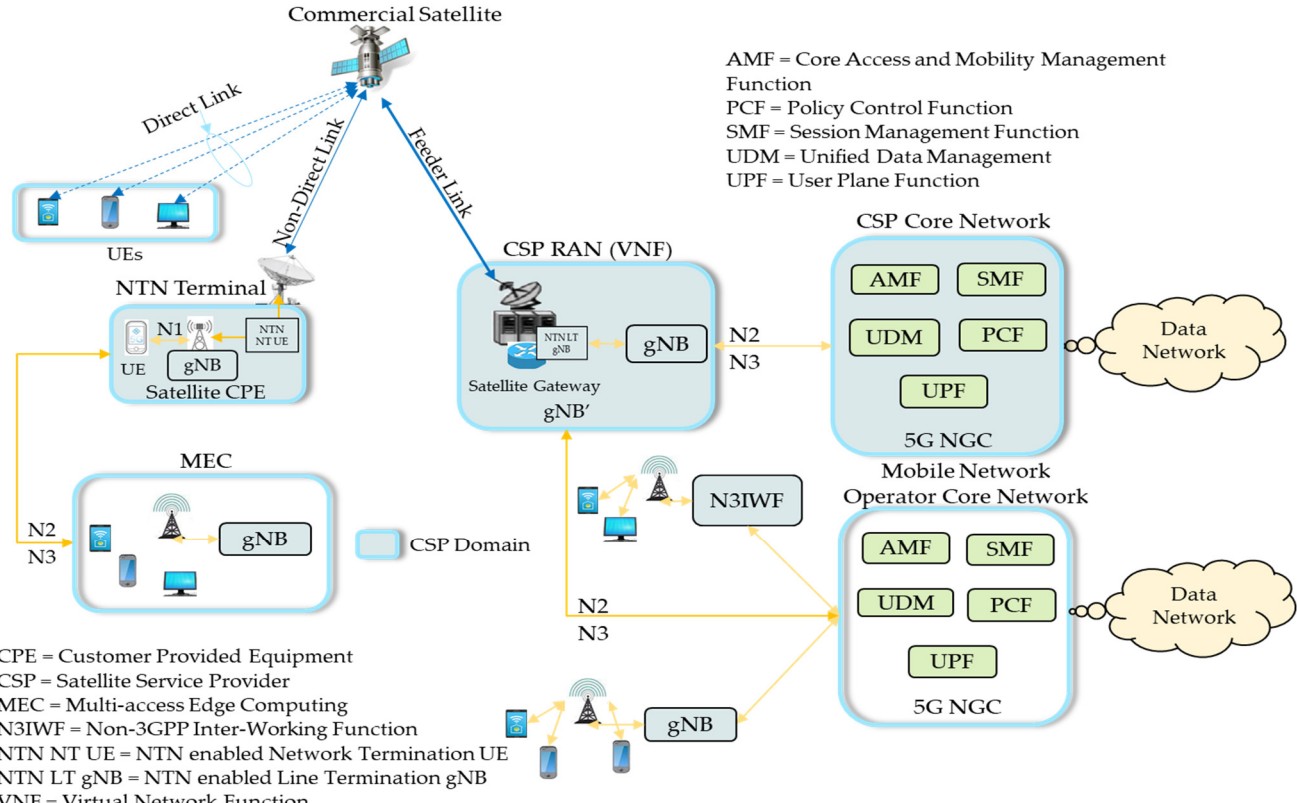

**Figure 14.** Top-level SATi5 architecture perspective for commercial applications.

The outlook for the potential SATis5 architecture described in Figure 14 is consistent with the roadmap discussed in Section 3.2. In the long-term, the integration of satellites and 5G-TN requires adopting the same 5G-NTN architecture recommended by 3GPP, including transmission, and switching technologies. The outlooked SATis5 architecture, shown in Figure 14, also requires modifying existing air interface protocols of the satellite access networks using existing 3GPP/5G-TN air interface protocols without changing the current recommended 3GPP/5G-UE air interface protocol stack. As mentioned earlier, the outlook perspective on commercial SATis5 can be extended to civilian SATis5.

### 6.2. Defense SATis5 Applications

The survey results captured in Section 5 indicate that the Aerospace defense industry is currently focusing on the development of SATis5 architectures that can address multi-domain defense operations. This domain includes but is not limited to (i) Integration of defense satellites with a 5G tactical gateway that enables JADC2 across all battlefield assets, (ii) Satellite-enabled 5G wireless network supporting operations and flight traffic management, (iii) Deploy secure 5G nodes at the tactical edge, and (iv) Integration of the GNSS with 5G networks enabling global positioning, navigation, and timing (PNT) capabilities.

In general, defense satellites represent a wide range of space-based systems, including military SATCOM satellites (e.g., US Wideband Global Satellite System), defense sensing

satellites (e.g., US Space Based Infrared System), and PNT satellites (e.g., US Global Positioning System). To achieve seamless multidomain defense operations enabling JADC2 across all battlefield assets, a unified enterprise SATis5 architecture requires integrating all existing defense satellites with 5G networks. This integration poses potential technical and programmatic challenges, including hardware/software interoperability, capabilities synchronization and alignment, and cost/schedule synchronization. This section provides our outlooks on potential SATis5 architectures for integrating existing and future MILSATCOM satellites with 5G networks.

The outlook for a defense SATis5 architecture that we envision is like Figure 14 with the commercial satellite being replaced by MILSATCOM satellites. Since MILSATCOM satellites employ TRANSEC mechanisms at PHY/Datalink/MAC layers, the commercial defense SATis5 architecture components are required to modify to address the TRANSEC mechanisms:

- MILSATCOM Satellite Payload: The payload's air interface should be modified to adapt to the 5G network's components, including NTN-LT-gNB, NTN-NT-UE, 5G-UEs, and 5G-NGC;
- CSP RAN (VNF): This CSP RAN can be replaced by a "Dedicated RAN" that allows the satellite gateway to process and manage the TRANSEC mechanisms associated with PHY/Datalink/MAC layers. The dedicated RAN will allow the satellite access to data networks through the 5G-NGC core network or mobile network operator core network. The NTN network termination gNB (NTN NT gNB) located at the satellite gateway should also be modified to (i) perform TRANSEC processing and management, and (ii) transport encapsulated control plane (CP) of the gNB;
- NTN Terminal: The Satellite CPE for the NTN Terminal can be replaced by "Dedicated NTN Terminal". The NTN Terminal should be modified to adapt to PHY/Datalink/MAC layers associated with TRANSEC mechanisms;
- 5G UEs and MILSATCOM Satellite User Terminals: The users' equipment and terminals should also be modified to adapt to the PHY/Datalink/MAC TRANSEC mechanisms associated with MILSATCOM satellites; and
- 5G-NGC: This should also be modified to manage the integrated SATis5 network resources effectively. The modified 5G-NGC is responsible for managing the use of communication resources optimally for both MILSATCOM and 5G networks.

In addition to the above modifications, other layers are also recommended for further investigation. For example, a mission optimizing orchestration layer, which is at Layer 8 or Layer 9 beyond the Application Layer of the standard OSI stack, will be needed for various JADC2 use cases for defense applications. Note that this orchestration layer provides "state-of-play" functions that can be supported by (i) 5G Mobile Edge Computing of Layer 0 and Layer 1, and (ii) Layer 2 and Layer 6 of cyber electronic warfare multi-access Edge Computing (EW MEC). The bottom-up approach for big data to build the state-of-play is based on the user plane function, unified data function, and policy control function of 5G technologies.

For US DoD, concerning the use of commercial technologies to meet defense space mission needs, the designated US DoD acquirers are required to make decisions based on the national interest incorporating cost, schedule, performance, and risk factors. Ref. [32] provides a framework for assisting the acquirers to assess a potential commercial solution for making informed decisions. The framework will allow for deciding whether a SATis5 solution can meet the national interest.

## 7. Discussion and Conclusions

Our survey results presented here have captured a wide range of SATis5 efforts that are fully complied with 3GPP/5G-NTN standards and specifications. Specific SATis5 architectures for eMBB and mMTC-NB-IoT were discussed. The outlooks on potential SATis5 architectures were provided. As discussed above, the outlooks were derived based on separate SATCOM enterprise perspectives. For the COMSATCOM perspective, the

outlook provided in Section 6.1 comprises 3GPP/SATis5 architecture standards and the roadmaps. The outlook for COMSATCOM can be extended to civilian SATCOM applications. From the defense SATCOM perspective concerning the use of MILSATCOM services integrated with 5G technologies, the outlook for long-term vision is the development of an emerging business model. This is an enterprise model that allows a cost-effective approach for accessing a virtual "unified integrated enterprise infrastructure" (IEI). The virtual IEI can support a wide range of user types. The IEI can be viewed as a unified enterprise infrastructure that virtually connects civilian, commercial, and defense SATCOM systems with all future global 5G networks. This IEI can also be envisioned as a "virtual" enterprise satellite network (EnVSaN) providing diverse communication services, including land, air, sea, and space, to support various user types. The EnVSaN would include all desired VNFs and related edge computing functions to support all global communications services for all user types. To achieve this long-term vision, the following investigations and studies are recommended:

- Investigate implementation approaches of the new 5G-NR air interfaces for existing and planned defense SATCOM payloads taking into consideration (i) Direct and Non-Direct access satellite links, (ii) satellite payload size-weight-power-and-cost (SWAP-C), (iii) MILSATOM users' QoS and Quality of Experience (QoE), (iv) satellite operation requirements and resource constraints, (v) 5G-UE SWAP-C, (vi) 5G users' QoS and QoE, and (vii) Battery life of 5G UE and satellite user terminal. The key technical area is the modification of existing 3GPP 5G-NR PHY/Datalink/MAC layers taking into account the long RTT associated with MILSATCOM satellites. These layers include waveforms, data formats, and data frames, time synchronization, RRC timer procedure, RAR time window size, and timing advance.
- Study implementing approaches for modifying the 5G-NR air interfaces for defense SATCOM satellites at NTN-NT-UE and NTN-LT/5G-gNB which can allow the SATCOM users and 5G users access to 5G-NGC and hence data networks. The key technical task is the incorporation of the existing ComPEMaC technique into the MILSATCOM payloads and 5G-Gateway based on the channel quality reported by either 5G-UE or MILSATCOM user terminals.
- Examin existing 5G-NGC's CP, including but not limited to AMF, SMF, UDM, PCF, and UPF, and propose control plan modifications taking into consideration of the newly modified 5G-NR air interfaces for defense SATCOM satellites.
- Develop innovative cost-effective business models which are derived based on the IEI perspective incorporating (i) Cost of IEI deployment, (ii) Cost of enterprise network maintenance and operations, (iii) Revenues for SATis5 services' providers and related vendors, (iv) Government cost saving for using SATis5 services and meeting defense mission needs, and (v) 5G and MILSATCOM users' QoS and QoE.

There is another study that is of interest to both commercial and defense users, this is the integration of GNSS with 5G networks. This study recommends focusing on providing PNT services to global 5G-UE and defense SATCOM user terminals in remote areas. Finally, the wireless communication industry is currently working on 6G addressing the proliferation of data-intensive applications that are not fully tackled by 5G systems due to limited bandwidth [33]. The emerging 6G technology offers new capabilities that eventually force the SATis5 transition to SATis6 architectures. When this happens, a similar study like this one is of interest to the commercial and defense SATCOM industry.

**Author Contributions:** Conceptualization, T.M.N. and K.D.P.; methodology, T.M.N.; formal analysis, T.M.N., K.D.P. and J.N.; investigation, T.M.N., K.D.P., J.N., G.C., C.H.L. and S.B.; writing—original draft preparation, T.M.N., K.D.P. and J.N.; writing—review and editing, T.M.N., K.D.P., J.N., G.C., C.H.L. and S.B. All authors have read and agreed to the published version of the manuscript.

**Funding:** This research received no external funding.

**Acknowledgments:** The authors express their sincere appreciation to Maria Rios of IFT for her careful review of our paper. The first author would like to express his appreciation for continuous support from the Center for Computational and Applied Mathematics at CSUF. He also wants to express his deepest appreciation to his wife, Thu-Hang Nguyen, for her unbounded patience and constant moral support during the process of writing this review.

**Conflicts of Interest:** The authors declare no conflict of interest. This work was carried out using our personal time and resources.

## Nomenclature

| Abbreviation | Description |
| --- | --- |
| 3GPP | The 3rd Generation Partnership Project |
| 4G | The fourth Generation wireless communication system |
| 5G | The fifth Generation wireless communication system |
| 5GIC | The 5G Internet Connection |
| 6G | The sixth Generation wireless communication system |
| ABR | Adaptive Bit Rate |
| AC-STE-iDirect | Avanti Communications and ST Engineering iDirect |
| AMF | Access and Mobility Management Function |
| ARTES | Advanced Research in Telecommunications Systems |
| ASTRA | Name of a satellite |
| BATF | Backhauling and Tower Feed |
| BS | Base Station |
| C2 | Command and Control |
| C4ISR | Command, Control, Communication Computer, Intelligence, Surveillance, Reconnaissance |
| CCAM | Center for Computational and Applied Mathematics |
| CDN | Content Distribution Networks |
| CMAF-CTE | Common Media Application Format-Chunked Transfer Encoding |
| COMSATCOM | Commercial Satellite |
| COTM | Comunication on the Move |
| COTS | Commercial of the Shelf |
| CP | Control Plane |
| CPE | Customer Provided Equipment |
| CSP | Satellite Service Provider |
| CT | Core Network and Protocol |
| CU | Central Unit |
| DU | Distributed Unit |
| eMBB | Enhanced Mobile Broadband |
| E2E | End-to-End |
| ESA | European Space Agency |
| ESTEC | European Space Research and Technology Centre |
| ETSI | European Telecommunications Standards Institute |
| EU | European |
| GEO | Geosynchronous Earth Orbit |
| GHz | Gega Hertz |
| GNSS | Global Navigation Satellite System |
| GSN | Ground Station Network |
| GW | Gateway |
| HAP | High Altitude Platform |
| HARQ | Hybrid Automatic Repeat Request |
| HYMP | Hybrid Multiplay |
| IEI | Integrated Enterprise Infrastructure |
| ISL | Inter-Satellite Link |
| ISR | Intelligence, Surveillance Reconnaissance |
| JADC2 | Joint All Domain Command and Control |
| LAP | Low Altitude Platform |
| LEO | Low Earth Orbit |



| | |
|---|---|
| LM | Lockheed Martin |
| LTE | Long-Term Evolution |
| M2M | Machine-to-Machine |
| MAC | Medium Access Layer |
| MEC | Multi-access Edge Computing |
| MEO | Medium Earth Orbit |
| MILSATOM | Military Satellite Communications |
| MPQUIC | Multipath version of the Quick UDP Internet Connections Protocol |
| NB | Narrow Band |
| NGC | Next-Generation Core Network |
| NGCN | Next-Generation Core Network |
| NR | New Radio |
| NTN | Non-Terrestrial Network |
| OFDM | Orthogonal Frequency Division Multiplexing |
| OSI | Open Systems Interconnection Model |
| PCF | Policy Control Function |
| PDCP | Packet Data Convergence Protocol |
| PER | Package Error Rate |
| PHY | Physical Layer |
| PL | Payload |
| PLMN | Public Land Mobile Network |
| PNT | Position, Navigation, Timing |
| RAN | Radio Access Network |
| RAR | Random-Access Response |
| RF | Radio Frequency |
| RN | Relay Node |
| RRC | Radio Resource Control |
| RTT | Round-Trip Time |
| SAT | Satellite |
| SATCOM | Satellite Communication |
| SD-WAN | Software-Defined Wide Area Networks |
| SES | Satellite Earth Stations and Systems |
| SMF | Session Management Function |
| SNR | Signal-to-Noise Ratio |
| SWAP | Size, Weight, and Power |
| TA | Timing Advance |
| TCP | Transport Control Protocol |
| THEF | Trucking and Head-End Feed |
| TN | Terrestrial Network |
| TR | Technical Report |
| TRANSEC | Transmission Security |
| UAV | Unmanned Aerial Vehicle |
| UDM | Unified Data Management |
| UDP | User Datagram Protocol |
| UE | User Equipment |
| Urllc | Ultra-Reliable and Ultralow Latency Communications |
| UPF | User Plane Function |
| UT | User Terminal |
| V2I | Vehicle-to-Infrastructure |
| V2V | Vehicle-to-Vehicle |
| VDC | Virtual Data Center |
| VLEO | Very Low Earth Orbit |
| VNF | Virtualization of Network Functions |
| VPN | Virtual Private Network |
| VSAT | Very Small Aperture Terminal |
| VSNF | Video-Segment scheduling Network Function |
| WAN | Wide Area Network |

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
