# Peer review of "A Review of SATis5: Perspectives on Commercial and Defense 5G SATCOM Integration"

_encyclopedia, doi:10.3390/encyclopedia2030087_

Round 1

Reviewer 1 Report

In this manuscript, the authors provide a concise review of review of past and existing works on 5G systems with a laser focus on 5G Satellite Integration (SATis5) for commercial and defense applications. The review manuscript is in general well-written, easy to follow and the references are appropriate, permitting a comprehensive view of the field. Please keep the same format of the fonts and the labels in all Figures, enhance their quality and improve their readability/aspect ratio (especially Figures 2, 4, 5 and Figure 14).

Author Response

We enhanced Figures 2, 4, 5 and Figure 14 for improved readability.

Many thanks.

Reviewer 2 Report

  • The authors have presented a review of potential SATis5 architectures for commercial and defense applications.
  • The article is very nicely organised and the literature review is carried out thoroughly. 
  • Summary using the tabular form while comparing with the existing work can give the readers a clear idea about the future perspectives of SATis5 architectures for the said applications.  

Author Response

Many thanks for the comments.

Our team has a special interest in 5G Satellite Integration (SATis5) architectures for commercial and defense applications, and this review paper focuses only on SATis5 architectures for these applications. The scope of our paper is not on the survey of 5G technologies, which is currently a hot topic for many survey papers and hence there are many existing works on this topic. Thus, writing a survey paper on this 5G technology topic required to provide a "Summary using the tabular form while comparing with the existing work." Concerning our contributions, our goal is not directed towards the identification of 5G technology gaps since many existing papers have thoroughly addressed these gaps. We place our focus on the gaps in SATis5 architectures for commercial and defense applications. We conducted a thorough search in the public domain and found only limited number of papers related to SATis5 architecture topic. Our revised paper has captured the work done by other researched on the SATis5 architectures' perspective in Section 3. Table 1 presented in Section 3 provided existing works related to SATis5 architectures in general. Table 4 of this section presented existing results for SATis5 architectures for commercial applications. For future perspective of SATis5 architectures for commercial and defense applications, Section 6 was revised to capture what was done by existing works and our key contributions. It is shown in Section 6 that only two papers addressed commercial SATis5 architectures and none for defense applications. The revised Section 6 and together with Section 7 showed that the key contribution of the paper is the SATis5 architecture outlooks for both commercial and defense applications.

Reviewer 3 Report

The paper presents a review of SATis5 focusing on the perspectives on commercial and defense 5G SATCOM integration. It is a timely, interesting subject and can bring relevant contributions to the literature. However, the manuscript needs major work before being processed further.

  • Generally, some sentences appear too long, making it quite challenging to understand the paper easily. Also, a few syntax errors and limited sentence constructions were identified, and these need to be revised accordingly.
  • A separate discussion section would greatly improve the manuscript, highlighting its key contributions. Before the paper's organization, the authors should state the key contributions of the review.
  • The problem being addressed in the review needs more clarification, and the review methodology needs to be well articulated for the benefit of the readers.
  • While I commend the current literature, it is expedient that the literature should be expanded accordingly to reflect the specific objectives of the current review. More recent and authoritative papers that are well structured and systematically in line with the stated objectives should be included in the review.
  • There is a need for a comprehensive Table summarizing the key literature review. Since the gaps in the related works are not well clarified in the current review, the authors should state the gaps in the existing literature that they have filled in the current review. The requested Table should show the gaps/limitations in the reviewed works and highlight clearly how each gap is addressed in the current study/review.
  • The first figure (Figure 1) would require further elaboration, especially on the working of the access network with the current core network.
  • Again, in my opinion, the research challenges and opportunities are not strong enough to attract expert readers. The authors should read popular reviews about the topic and summarize more potential.
  • What is the major limitation of the current survey? What do you think researchers should do to improve the current survey?
  • The future work is unclear in conclusion, so I encourage the authors to revise its future scope appropriately.
  • Although I think that the paper has provided a good overview of the issue being addressed, I still wonder how things are going in Europe, Africa, and other regions of the world. Including examples (if any) may be of help.
  • I must commend the diagrams that make the paper more exciting to read, but I suggest a few more explanations of each diagram.
  • Overall, the reference is not well spread and grossly insufficient for a review paper. The authors should add the most recent, quality, and authoritative papers related to the current topic and the major claims in the paper. I suggest that you add this paper to the review: https://doi.org/10.3390/s21051709
  • Since several acronyms are used in the paper, adding a list of abbreviations at the end of the conclusion section would be nice.
  • It would be nice to provide the official email addresses of all authors. Also, check the affiliations and add the correspondence.

Author Response

  1. Generally, some sentences appear too long, making it quite challenging to understand the paper easily. Also, a few syntax errors and limited sentence constructions were identified, and these need to be revised accordingly.

Authors’ Response: many thanks for pointing out this deficiency. Since the paper is written by multiple authors, this is unavoidable. It should be nice if the reviewer pointed out those sentences for us to fix them. However, our team carefully reviewed the paper, looked for those sentences, and corrected them.

  1. A separate discussion section would greatly improve the manuscript, highlighting its key contributions. Before the paper's organization, the authors should state the key contributions of the review.

Authors’ Response: The paper is organized to meet (i) the page constraint, and (ii) our objectives. The discussion is adequately combined into Section 7, hence there is no need for a separate discussion section. Concerning our contributions for this paper, we make sure that these contributions are clearly presented in the introduction section.       

Since our team has special interest in 5G Satellite Integration (SATis5) architectures for commercial and defense applications, this review paper has a laser focus on SATis5 for these applications. We make sure that this objective is clearly stated in our abstract and the introduction section. Concerning our contributions, this paper provides our outlook perspectives on potential SATis5 architectures for both commercial and defense applications. As discussed in Section 6.1, our outlooks for commercial SATis5 architectures are consistent with other researchers. Concerning defense SATis5 outlooks, Section 6.2 pointed out that (i) there are no available SATis5 architecture results for defense applications, and (ii) our outlook for the defense applications is derived from our own survey on recent and existing defense SATis5 testbeds and projects. Our SATis5 architecture vision for defense is very similar to the commercial view but with new issues and challenges. Our team identifies these issues/challenges and corresponding studies addressing these new issues/challenges. We make sure that these contributions are clearly stated in the revised version.

  1. The problem being addressed in the review needs more clarification, and the review methodology needs to be well articulated for the benefit of the readers.

Authors’ Response: This comment demonstrated that this reviewer is not clear on our team’s objectives and review approach to achieve our objectives. It’s our fault that we have not made them clear to all reviewers/readers. We make sure that these areas are clear in the revised version.

In terms of the objectives of this paper, we have addressed them in the above Comment #2. Concerning our review approach (referred to this reviewer as review methodology), our team used a holistic survey approach allowing us to gain an in-depth understanding on recent and current SATis5 works, and from this understanding our team derived potential outlooks on SATis5 architectures and related issues and challenges for commercial and defense applications. Thus, the objective of this review paper is to two-fold: (1) Provide outlook perspectives on potential SATis5 architectures for defense and commercial applications, (2) Gain in depth understanding on issues and challenges associated with our anticipated outlook on SATis5 architectures and generate a new set of studies to address these issues/challenges. Our team make an effort to ensure that this is clear in the next version.     

  1. While I commend the current literature, it is expedient that the literature should be expanded accordingly to reflect the specific objectives of the current review. More recent and authoritative papers that are well structured and systematically in line with the stated objectives should be included in the review.

Authors’ Response: This comment is the same as Comments #2 and #3. Based on our responses to the above comments and the works described in this paper, our team strongly feels that our holistic survey approach is sounded and consistent with the objectives set forth. Using this approach, we have thoroughly conducted a survey covering all past and recent authoritative works on SATis5 architectures for commercial and defense applications. The survey results allowed us to achieve our objectives. Remember that our paper does not intent to provide a comprehensive survey on 5G (or 4G or 6G or the evolution from 4G to 5G) technologies and their applications. We survey existing results on 5G technologies and their applications with a laser focus on SATis5 architectures and roadmaps for commercial and defense applications. Again, we make sure that these are clear in the next revision.

  1. There is a need for a comprehensive Table summarizing the key literature review. Since the gaps in the related works are not well clarified in the current review, the authors should state the gaps in the existing literature that they have filled in the current review. The requested Table should show the gaps/limitations in the reviewed works and highlight clearly how each gap is addressed in the current study/review.

Authors’ Response: Again, this very similar to Comments 2, 3, and 4 above. Please see our responses above.

Our goal is not on the identification of gaps in the previous survey works and fill the gaps. Many existing papers have thoroughly addressed these gaps. We recognize the gaps of SATis5 architectures for commercial and defense applications. Our paper addresses these SATis5 architectures gaps.    

  1. The first figure (Figure 1) would require further elaboration, especially on the working of the access network with the current core network.

Authors’ Response: As mentioned above, the focus of our paper is NOT to provide the details of Figure 1, since we provided enough references for the readers to dig into details of Figure 1 if needed. We show Figure 1 to introduce all 5G functions so that the readers can relate some of these key 5G functions will be essential for the identification SATis5 issues and challenges for commercial and defense applications.

  1. Again, in my opinion, the research challenges and opportunities are not strong enough to attract expert readers. The authors should read popular reviews about the topic and summarize more potential.

Authors’ Response: We are not quite sure what expert readers that this reviewer is trying to convey. Since the paper focused on satellite integration with 5G terrestrial network for commercial and defense applications, we will say that, for the expert readers, they will be interested on (i) the issues and challenges associated SATis5 architectures for commercial and defense applications, and (ii) how to develop future research projects to address these issues/challenges. Thus, our intent is to attract expert readers in the commercial and defense SATCOM applications not necessary 4G/5G/6G experts. However, 4G/5G/6G experts might want to read our paper to find out the issues and challenges associated with SATis5 for defense applications.  

  1. What is the major limitation of the current survey? What do you think researchers should do to improve the current survey?

Authors’ Response: Like the above comments. Please see our responses above.

We recognized there are limited survey papers on SATis5 architectures for commercial application. We think our team is a unique group to address these application areas.   

  1. The future work is unclear in conclusion, so I encourage the authors to revise its future scope appropriately.

Authors’ Response: Many thanks for pointing this out. Our team make an effort to improve the clarity of future work.  

  1. Although I think that the paper has provided a good overview of the issue being addressed, I still wonder how things are going in Europe, Africa, and other regions of the world. Including examples (if any) may be of help.

Authors’ Response: Please see Section 4 for discussion of SATis5 in Europe that is within the scope of this paper. If you find any papers that are within the scope of our paper, please let us know. Many thanks in advance.

  1. I must commend the diagrams that make the paper more exciting to read, but I suggest a few more explanations of each diagram.

Authors’ Response: As pointed out earlier, the scope of this paper is on SATis5 architectures for commercial and defense applications, it is a large scope for a 24-page paper. We only present enough information to convey key ideas associated with the figures allowing readers with general background with SATCOM systems can understand quickly. However, our team make an extra effort to enhance the clarity of the explanations of each diagram.

  1. Overall, the reference is not well spread and grossly insufficient for a review paper. The authors should add the most recent, quality, and authoritative papers related to the current topic and the major claims in the paper. I suggest that you add this paper to the review: https://doi.org/10.3390/s21051709

Authors’ Response: This comment clearly shown that the reviewer did not understand the intent of the paper! Of course, it is our fault that our intent is not clearly conveyed in a 24-page. Again, this is not a comprehensive review of 5G technologies and their applications. Thus, we respectfully disagree with the above assessment. As mentioned above, our team make an extra effort to convey the intent of our work.

Many thanks for suggesting the above paper as a reference. The reference you provided is very good for a comprehensive view on wireless communication evolution from 1G to 6G along with future 6G requirements and associated use cases. Clearly, this recommended reference is not in line with the work presented in our paper (we focus on SATis5). However, we mention this reference in Section 7 where we discuss the future work for long term vision where 6G becomes more mature compelling a transition from SATis5 to SATis6 architectures.

  1. Since several acronyms are used in the paper, adding a list of abbreviations at the end of the conclusion section would be nice.

Authors’ Response: Yes, we agree. This list can be generated easily by the publisher using automated publication tool (if needed).

  1. It would be nice to provide the official email addresses of all authors. Also, check the affiliations and add the correspondence.

Authors’ Response: Yes, we agree. This can be done easily if the publisher requests the information.

Round 2

Reviewer 1 Report

The authors have significantly enhanced the quality of their manuscript based on the reviewers' comments, which I now recommend for publication.

Author Response

No comment from Reviewer 1 for the second round.

Reviewer 2 Report

- The paper is a in a good condition to be accepted for publication. 

Author Response

No comment from Reviewer 2 for the second round.

Reviewer 3 Report

I suggest that the list of abbreviation is required.

Author Response

The second revised version had incorporated the list of abbreviations per Reviewer-3's suggestion.

Round 3

Reviewer 3 Report

The authors have addressed my comments.